# Energy coupling and stoichiometry of Zn$^{2+}$/H$^+$ antiport by the prokaryotic cation diffusion facilitator YiiP

Adel Hussein[1†], Shujie Fan[2†], Maria Lopez-Redondo[1], Ian Kenney[2], Xihui Zhang[1], Oliver Beckstein[2]*, David L Stokes[1]*

[1]Department of Biochemistry and Molecular Pharmacology, NYU School of Medicine, New York, United States; [2]Department of Physics, Arizona State University, Tempe, United States

**Abstract** YiiP from Shewanella oneidensis is a prokaryotic Zn$^{2+}$/H$^+$ antiporter that serves as a model for the Cation Diffusion Facilitator (CDF) superfamily, members of which are generally responsible for homeostasis of transition metal ions. Previous studies of YiiP as well as related CDF transporters have established a homodimeric architecture and the presence of three distinct Zn$^{2+}$ binding sites named A, B, and C. In this study, we use cryo-EM, microscale thermophoresis and molecular dynamics simulations to address the structural and functional roles of individual sites as well as the interplay between Zn$^{2+}$ binding and protonation. Structural studies indicate that site C in the cytoplasmic domain is primarily responsible for stabilizing the dimer and that site B at the cytoplasmic membrane surface controls the structural transition from an inward facing conformation to an occluded conformation. Binding data show that intramembrane site A, which is directly responsible for transport, has a dramatic pH dependence consistent with coupling to the proton motive force. A comprehensive thermodynamic model encompassing Zn$^{2+}$ binding and protonation states of individual residues indicates a transport stoichiometry of 1 Zn$^{2+}$ to 2–3 H$^+$ depending on the external pH. This stoichiometry would be favorable in a physiological context, allowing the cell to use the proton gradient as well as the membrane potential to drive the export of Zn$^{2+}$.

*For correspondence:
obeckste@asu.edu (OB);
stokes@nyu.edu (DLS)

†These authors contributed
equally to this work

Competing interest: The authors
declare that no competing
interests exist.

Reviewing Editor: Merritt
Maduke, Stanford University
School of Medicine, United
States

## eLife assessment

This **important** and elegant study uses experimental structural data, ion affinity measurements, and computational methods to provide insight into the thermodynamic landscape of cation transporters of the Cation Diffusion Facilitator (CDF) superfamily, together with a detailed structural investigation of the role of the three zinc(II) binding sites of the YiiP family member. Overall, the support for the proposed transport cycle of YiiP is **compelling**. This work will be of interest to biologists and biophysics who work with membrane transporters.

## Introduction

YiiP is a bacterial Zn$^{2+}$/H$^+$ antiporter and a well-characterized representative of the Cation Diffusion Facilitator (CDF) superfamily. Members of this family play important roles in homeostasis of transition metal ions such as Zn$^{2+}$, Mn$^{2+}$, Co$^{2+}$, and Fe$^{2+}$ (*Montanini et al., 2007*). Mn$^{2+}$ transporters from this superfamily are prevalent in plants, where this metal ion is essential for oxygen generation by photosystem II as well as for a variety of other enzymatic functions (*Alejandro et al., 2020*). Zn$^{2+}$ CDF transporters are widespread in organisms from all kingdoms of life reflecting a large influence of this ion on cell biology. An estimated 10% of proteins employ Zn$^{2+}$ either as a catalytic co-factor (e.g.,

carbonic anhydrase and cytochrome C oxidase) or as a structural element (e.g., Zn-finger transcription factors and steroid receptors; *Maret, 2013*). In mammals, high concentrations are found in a variety of intracellular vesicles that are key players in the immune system, synaptic transmission, insulin trafficking, and fertilization (*Kambe et al., 2015*). $Zn^{2+}$ figures in host-pathogen interactions, with the host attempting to either deprive or poison pathogens residing in endosomes (*Lonergan and Skaar, 2019*). Despite its prevalence, the bulk of intracellular $Zn^{2+}$ is bound to protein, with vanishingly small ($<10^{-12}$ M) concentrations of free $Zn^{2+}$ in the cytoplasm (*Liang et al., 2016*; *Outten and O'Halloran, 2001*). Homeostasis is maintained by CDF transporters, which generally export $Zn^{2+}$ from the cytoplasm, as well as members of the Zinc-regulated or Iron-regulated transport Proteins (ZIP), P-type ATPase and ATP-binding cassette (ABC) superfamilies (*Yin et al., 2023*).

YiiP serves as a model for studying structural and mechanistic properties of CDF transporters. Structures of homologs from both *Escherichia coli* and *Shewanella oneidensis* in different conformational states provide a framework for describing the archetypal alternating access mechanism (*Lopez-Redondo et al., 2021*; *Lopez-Redondo et al., 2018*; *Lu et al., 2009*; *Lu and Fu, 2007*). Molecular Dynamics has been used to study dynamics (*Sala et al., 2019*), to define the transport pathway (*Sharma and Merz, 2022*) and, together with in vitro binding assays, to characterize the $K_d$ and pKa of Zn binding sites. Our understanding of CDF transporters is amplified by recent MD simulations of Znt2 (*Golan et al., 2018*; *Golan et al., 2019*), cryo-EM structures of Znt8 (*Daniels et al., 2020*; *Xue et al., 2020*; *Zhang et al., 2023*) and Znt7 (*Bui et al., 2023*), as well as X-ray structures of isolated cytoplasmic domains from a number of different species (*Udagedara et al., 2020*). Together, these studies establish characteristics of $Zn^{2+}$ binding sites, a conserved homodimeric architecture, and the nature of inward-facing (IF) and outward-facing (OF) states that presumably characterize all CDF transporters.

Despite this extensive work, there is uncertainty about key mechanistic questions that we have sought to address in the current work. Our primary goal was to obtain experimental evidence for the individual roles of the three $Zn^{2+}$ binding sites found on each YiiP protomer (*Cotrim et al., 2019*). Site A is within the transmembrane domain (TMD) and features three Asp and one His residues; it is alternately exposed to the cytoplasm or to the periplasm in IF and OF states, respectively, and is thus directly responsible for transport of ions across the membrane. Although each molecule composing the dimer appears to have an independent transport pathway, there is unresolved potential for cooperativity. Site C within the C-terminal domain (CTD) is a binuclear site featuring four His and two Asp residues; one of these sites (C1) is well conserved whereas the other (C2) is not (*Parsons et al., 2018*). Site B is on the loop between transmembrane helices 2 and 3 (TM2-TM3) featuring two His and one Asp residue. Despite a lack of sequence homology for sites B and C2, the structure of Znt8 showed Zn ions at similar locations of the tertiary fold, but bound at different locations in the linear sequence, suggesting that these non-conserved, auxiliary sites may have functional significance (*Xue et al., 2020*).

Another goal was to address the role of protons in the transport process. There is ample evidence supporting $Zn^{2+}/H^+$ antiport. In vitro assays with Znt1, Znt2, ZitB and CzcD showed pH dependence of transport (*Chao and Fu, 2004a*; *Cotrim et al., 2021*; *Guffanti et al., 2002*; *Shusterman et al., 2014*). Binding studies with Znt8 (*Zhang et al., 2023*) as well as computational studies with Znt2 (*Golan et al., 2019*) and YiiP (*Sharma and Merz, 2022*) showed an interplay between $Zn^{2+}$ and $H^+$ binding at site A. However, there are discrepancies in stoichiometry, with work on ZitB (*Chao and Fu, 2004a*) and YiiP (*Chao and Fu, 2004b*) supporting a 1:1 exchange of $Zn^{2+}$ and $H^+$, whereas studies of CzcD (*Guffanti et al., 2002*) and Znt2 suggest 1:2 (*Golan et al., 2019*). There is also recent evidence for $Zn^{2+}/Ca^{2+}$ antiport by Znt1 in neurons (*Gottesman et al., 2022*).

For the current study, we have generated mutants of YiiP and have used cryo-EM and microscale thermophoresis (MST) to measure the pH dependence and structural effects of $Zn^{2+}$ binding at each of the three sites. In addition, we have used molecular dynamics (MD) simulations together with the experimental MST data to deduce the pKa of residues at these sites and to address the stoichiometry of transport. From these data, we conclude that $Zn^{2+}$ binding at site C is responsible for the integrity of the homodimer. Release of $Zn^{2+}$ from site B triggers a conformational change in which the transport site A becomes occluded, suggesting a potential relay of $Zn^{2+}$ between these two sites. Occlusion of only one protomer breaks the symmetry of the dimer and suggests that the transport process is not cooperative. In addition, binding affinity at site A displays a dramatic pH dependence, which can be explained by the protonation of 2 or 3 of the residues comprising this site. A corresponding $Zn^{2+}/H^+$

antiport stoichiometry of 1:2 or 1:3, depending on pH, is consistent with energetic coupling of $Zn^{2+}$ export to the proton-motive force in a physiological setting.

## Results

### Mutants used to study the structural effects of $Zn^{2+}$ binding

In order to study effects of $Zn^{2+}$ binding at individual sites, we produced mutants of YiiP from *S. oneidensis*. The D51A mutation was introduced to preclude binding at site A and D70A to preclude binding at site B. For site C, we mutated Asp287, because it bridges the two $Zn^{2+}$ ions at that site and we anticipated that D287A would therefore eliminate both ions. The wild-type (WT) protein studied in previous work (*Coudray et al., 2013*; *Lopez-Redondo et al., 2021*; *Lopez-Redondo et al., 2018*) served as a positive control. For each of these constructs, protein was expressed in *E. coli*, solubilized in decyl-β-D-maltoside and purified by affinity chromatography. To ensure binding of $Zn^{2+}$ rather than other metal ions picked up during growth and initial purification (e.g., $Ni^{2+}$), we incubated the protein overnight with chelators and then added 250 µM $Zn_2SO_4$ prior to the final purification step. As in previous work (*Lopez-Redondo et al., 2021*), we used an Fab antibody fragment to produce a larger complex amenable to cryo-EM analysis: 165 kDa comprising the YiiP homodimer (65 kDa) bound to two Fab's (50 kDa each).

### Site A: D51A mutation

The structure of the D51A complex was solved at 3.6 Å resolution (*Figure 1B*, *Figure 1—figure supplements 1 and 2*, *Table 1*) revealing an architecture very similar to untreated, WT YiiP (PDB code 7KZZ) that represents the IF, holo state (*Lopez-Redondo et al., 2021*). Specifically, YiiP formed a twofold symmetric homodimer with Fab molecules bound near the C-terminus of each CTD. Density for $Zn^{2+}$ ions is clearly visible at sites B and C (*Figure 1—figure supplement 1E and F*). At site A, however, this structure lacked density (*Figure 1—figure supplement 1D*), confirming that the D51A mutation effectively eliminated binding at this site.

For comparison, we solved the structure from the WT construct that was similarly loaded with $Zn^{2+}$. As expected, this 3.8 Å resolution structure (*Figure 1A*, *Figure 1—figure supplement 2*) is indistinguishable from 7KZZ, which is also from the WT construct but not treated with chelators or explicitly loaded with $Zn^{2+}$ (RMSD 1.07 Å for all 562 Cα atoms), Despite overall similarity, comparison with the D51A structure showed movements of TM1, TM4 and TM5 (*Figure 2a*), which resulted in a notably elevated RMSD for this region: 4.7 Å based on the corresponding 200 Cα atoms compared to 0.69 Å for the 352 residues composing the rest of the YiiP dimer. The cytoplasmic end of TM5 had the largest differences. Specifically, a kink is introduced near His155 and the TM4-TM5 loop is disordered (*Figure 1—figure supplement 1D*), suggesting a flexibility of this region in the absence of $Zn^{2+}$ at site A.

### Site B: D70A mutation

An initial structure of the D70A construct revealed a global conformational change reminiscent of the previously solved WT, apo state (PDB 7KZX) produced by removal of $Zn^{2+}$ with chelators (*Lopez-Redondo et al., 2021*). The most salient feature of this conformation is a kink between TMD and CTD, which disrupts the overall two-fold symmetry of the complex. During image processing, it became evident that there were in fact multiple conformations in the dataset from D70A, which ultimately generated two distinct structures at 4.0 and 3.9 Å resolution (*Figure 1C*, *Figure 1—figure supplement 3*). Unlike the WT, apo structure (7KZX), both of these new structures revealed densities at sites A and C (*Figure 1—figure supplement 1G,I,J and L*), consistent with $Zn^{2+}$ binding to intact sites. However, the two structures differed both in the conformation of the TMD and in the orientation of the CTD. One of these structures, termed D70A_sym, has a TMD displaying local two-fold symmetry, in which both protomers adopt the IF state (*Figure 2B*). Like the WT, apo structure, the TM2/TM3 loops, which harbor Asp70 and site B, are disordered. In the other structure, termed D70A_asym, one TMD adopts the IF state with a disordered TM2/TM3 loop, whereas the other TMD adopts a novel conformation in which the TM2/TM3 loop extends ~17 Å away from the membrane to interact with H1 and β1 elements of the CTD from the opposing protomer (*Figure 2C*, *Figure 2—figure supplement 1B*). In addition to reconfiguring this loop, there are movements in TM1,2,4,5 (*Figure 2D*) that close

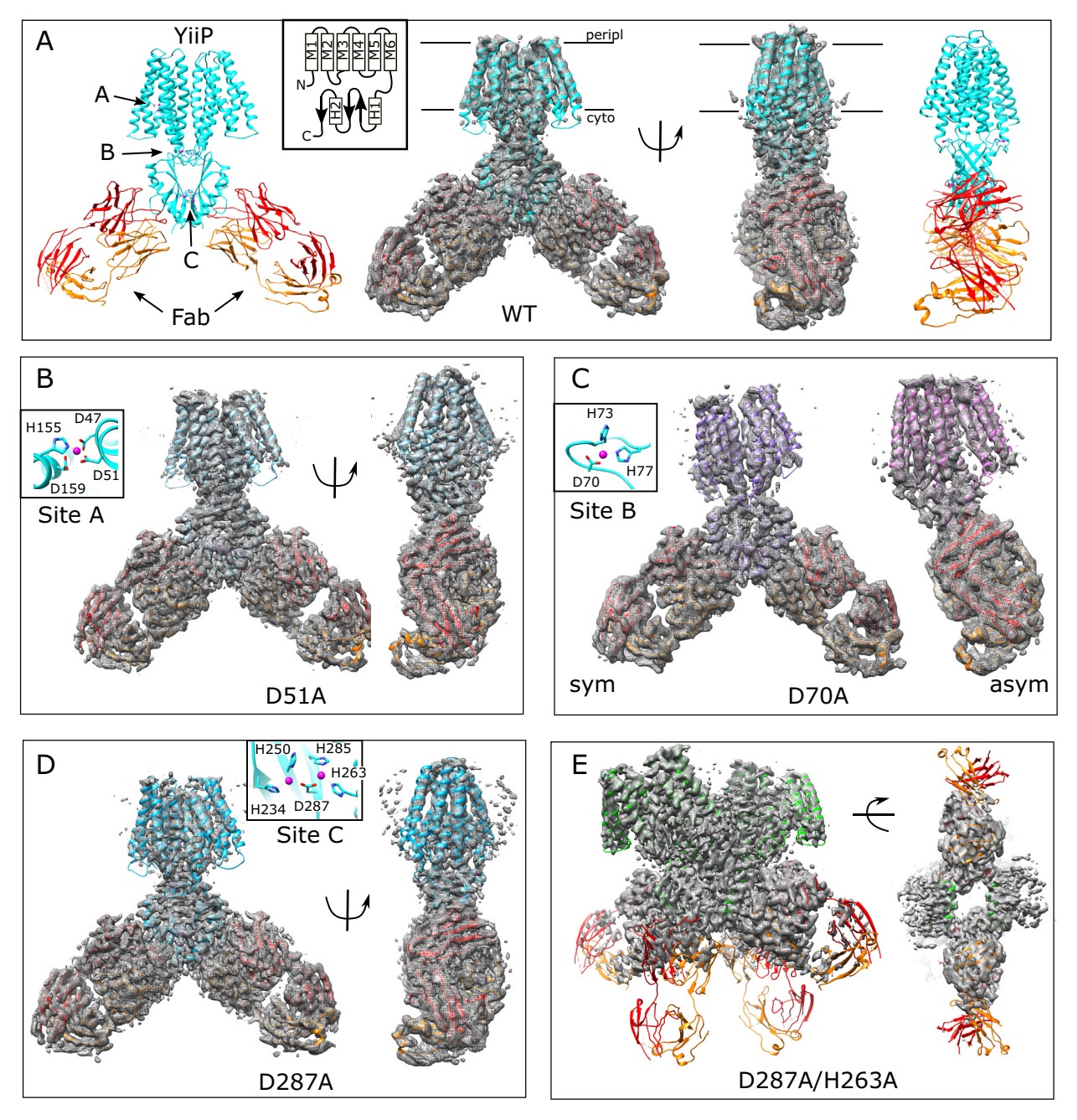

**Figure 1.** Overview of the cryo-EM structures. Density maps and corresponding atomic models are shown for each mutant. Fab molecules are colored orange (light chain) and red (heavy chain) with YiiP colored in cyan, blue, purple and green, depending on the mutant. The homo-dimers adopt C2 symmetry for WT (**A**), D51A (**B**) and D287A (**D**) mutants, but a bend between TMD and CTD break this symmetry for D70A (**C**). The D287A/H263A mutant (**E**) forms a dimer of dimers in which the Fab molecules are rather disordered. Two conformations were observed for D70A, both of which are shown in panel C: D70A_sym on the left and D70A_asym on the right. Location $Zn^{2+}$ binding sites, membrane boundaries as well as the topology of the YiiP protomer (inset) are shown in panel A; rectangles and arrows represent α-helices and β-sheets, respectively. Insets in panels B-D show the coordination geometry at the individual sites. Although the resolution was not always sufficient to uniquely define the side chain orientations, the maps are fully consistent with coordination geometry originally defined in the higher resolution X-ray structure (*Lu et al., 2009*).

The online version of this article includes the following figure supplement(s) for figure 1:

**Figure supplement 1.** Density at $Zn^{2+}$ sites in the cryo-EM maps.

**Figure supplement 2.** Determination of WT and D51A structures by cryo-EM.

*Figure 1 continued on next page*

*Figure 1 continued*

**Figure supplement 3.** Determination of D70A structures by cryo-EM.

**Figure supplement 4.** Determination of D287A and D287A/H263A structures by cryo-EM.

**Table 1.** Structure determination of YiiP mutants.

| Data set | WT | Site A D51A | Site B* D70A asymTMD | Site B* D70A symTMD | Site C D287A | Site C[2] D287A/H263A |
|---|---|---|---|---|---|---|
| **Deposition** | | | | | | |
| PDB | 8F6E | 8F6F | 8F6H | 8F6I | 8F6J | 8F6K |
| EMDB | EMD-28881 | EMD-28882 | EMD-28883 | EMD-28884 | EMD-28885 | EMD-28886 |
| **Data collection and processing** | | | | | | |
| Magnification | 81,000 | 81,000 | 81,000 | 81,000 | 81,000 | 81,000 |
| Voltage (kV) | 300 | 300 | 300 | 300 | 300 | 300 |
| Electron exposure ($e^-/Å^2$) | 50 | 50 | 50 | 50 | 50 | 50 |
| Defocus range (μm) | 1.0–3.0 | 0.75–2.75 | 0.7–3.0 | 0.7–3.0 | 0.7–3.0 | 0.7–2.5 |
| Pixel size (Å) | 1.068 | 1.079 | 1.079 | 1.079 | 1.079 | 1.079 |
| Symmetry imposed | C2 | C2 | C1 | C1 | C2 | C2 |
| Initial particle images (no.) | 3,058,414 | 3,672,562 | 1,702,119 | 1,702,119 | 1,664,097 | 2,982,749 |
| Final particle images (no.) | 536,206 | 196,484 | 188,414 | 182,413 | 252,599 | 300,844 |
| Map resolution FSC threshold (Å) | 3.78 | 3.63 | 3.93 | 4.03 | 3.68 | 3.46 |
| B factor ($Å^2$) | 0.143 | 0.143 | 0.143 | 0.143 | 0.143 | 0.143 |
| Resolution range (Å) | 200.5 | 151.0 | 146.1 | 126.0 | 150.9 | 135.2 |
| | 3.0–5.5 | 3.0–5.2 | 3.4–5.8 | 3.4–6.9 | 3.0–5.5 | 3.0–5.2 |
| **Model Refinement** | | | | | | |
| Model composition Non-hydrogen atoms | 10478 | 10348 | 10421 | 10377 | 10472 | 8714 |
| Protein residues | 1366 | 1344 | 1358 | 1351 | 1366 | 1132 |
| Ligands | 8 | 6 | 6 | 6 | 8 | 4 |
| RMS deviations Bond lengths (Å) | 0.003 | 0.003 | 0.002 | 0.002 | 0.003 | 0.002 |
| Bond angles (°) | 0.557 | 0.535 | 0.509 | 0.517 | 0.536 | 0.499 |
| Validation MolProbity score Clashscore Rotamer outliers (%) | 1.70 | 1.68 | 1.72 | 1.75 | 1.77 | 1.96 |
| | 5.44 | 6.31 | 5.78 | 7.11 | 6.37 | 10.28 |
| CaBLAM outliers (%) | 0.00 | 0.09 | 0.00 | 0.00 | 0.00 | 0.00 |
| | 2.66 | 3.05 | 2.60 | 3.40 | 3.64 | 4.21 |
| Rama-Z score | 0.77 | 0.37 | 0.05 | 0.20 | 0.50 | 0.45 |
| Ramachandran plot Favored (%) | 94.19 | 95.27 | 93.99 | 94.86 | 93.59 | 93.51 |
| Allowed (%) | 5.44 | 4.57 | 5.86 | 4.99 | 6.04 | 5.96 |
| Disallowed (%) | 0.37 | 0.15 | 0.15 | 0.15 | 0.37 | 0.53 |
| Model vs. Data CC (mask) | 0.75 | 0.83 | 0.79 | 0.80 | 0.81 | 0.73 |

*Both D70A structures arose from the same set of micrographs and initial particle picks. [2] Fab molecules were not included for the refinement of D287A/H263A due to poor density in this region of the map.

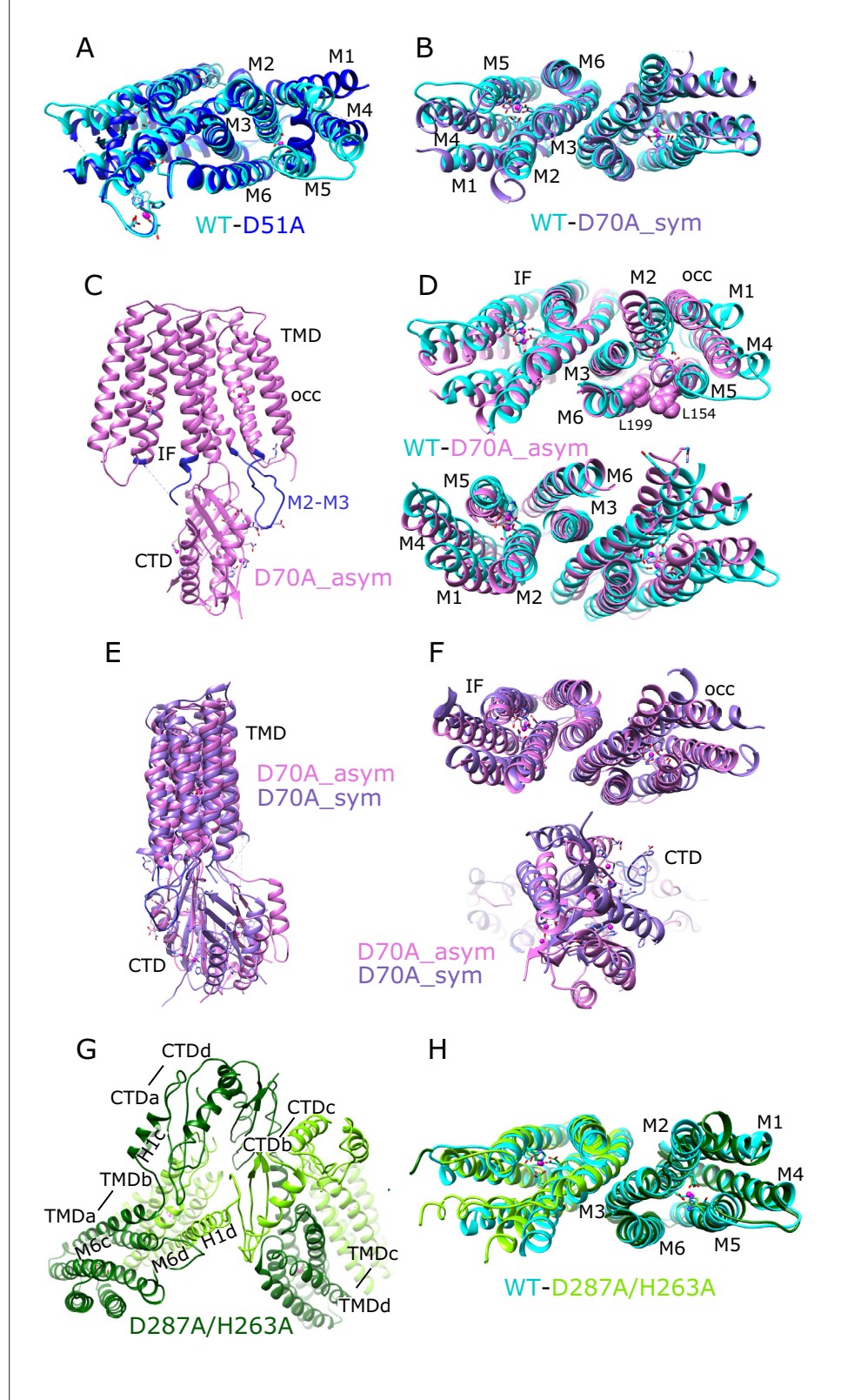

**Figure 2.** Comparisons of atomic models. (**A**) Overlay of the TMD for WT (cyan) and D51A (blue) structures. This view is from the cytoplasmic side of the membrane, tilted slightly to show site A in protomer A. Core helices (**M2, M3, M6**) are well aligned, but there are substantial displacements of the peripheral helices (**M1, M4, M5**), presumably due to lack of $Zn^{2+}$ binding at site A serving to bridge M2 and M5. (**B**) Overlay of the TMD for WT

*Figure 2 continued*

(cyan) and D70A_sym (purple) shows a very similar configuration of helices and an intact dimeric interface. This view is directly along the two-fold axis from the cytoplasmic side of the membrane. (**C**) Structure of D70A_asym viewed along the membrane plane showing the asymmetry between the two TMD's. The protomer on the left (chain B) adopts an IF conformation, whereas the protomer on the right (chain A) adopts a novel occluded conformation that includes a reconfigured TM2/TM3 loop (blue) making a novel interaction with the CTD. (**D**) Overlay of the TMD for WT (cyan) and D70A_asym (plum) showing the occluded conformation adopted by protomer A (top) and the IF conformation adopted by protomer B (bottom). L199 and L154 (spheres) make van der Waals interactions that appear to stabilize the occluded conformation. These views are from the cytoplasmic side of the membrane, slightly tilted to show the respective protomers. (**E–F**) Overlays of D70A_asym (plum) and D70A_sym (purple) structures after alignment of core helices TM3 and TM6. Structures are viewed parallel to the membrane plane in E and along the two-fold axis from the cytoplasmic side of the membrane in F. A significant shift in the position of the CTD is apparent in E and bottom panel in F. Despite this shift and substantial conformational changes in the occluded protomer A, the dimer interface in the TMD (TM3 and TM6) is well aligned (F, top panel). (**G**) Domain swapped dimer of dimers adopted by the D287A/H263A construct. Dimerization of TMD's involves interaction between one dark-green and one light-green molecule (e.g., TMDa and TMDb, where 'a' and 'b' refer to chain ID), whereas dimerization of CTD's involves interaction between either two dark-green or two light-green molecules (e.g., CTDa and CTDd). The linker between M6 and the CTD adopts a long straight helix in chains b and d, but remains an unstructured loop in chains a and c. This view is from the cytoplasm looking toward the membrane surface. (**H**) Overlay of the TMD for WT (cyan) and D287A/H263A structures viewed along the dimer axis from the cytoplasmic side of the membrane shows a good match, indicating that disruption of site C affects mainly the configuration of the CTD.

The online version of this article includes the following source data and figure supplement(s) for figure 2:

**Figure supplement 1.** Interactions between the TM2/TM3 loop and the CTD.

**Figure supplement 2.** Water accessibility of site A.

**Figure supplement 2—source data 1.** Source data plotted in panel C.

off site A from the cytoplasm. This occlusion is documented by the considerably smaller radius of the cavity leading to this site (*Figure 2—figure supplement 2*).

Despite these changes, local dimeric elements from the D70A structures superimpose closely onto the corresponding elements from the WT, holo structure. In particular, alignment of CTD's produces low RMSD's of 0.82 Å and 1.08 Å relative to D70A_sym and D70A_asym, respectively (168 Cα atoms). The core helices mediating the TMD dimer interface (TM3/6) also align closely with WT with RMSD's of 0.79 Å and 1.22 Å for D70A_sym and D70A_asym (106 Cαatoms, *Figure 2B and D*). For D70A_sym, the entire TMD is generally consistent with the WT, holo structure (*Figure 2B*), although slight displacement of TM1,4,5 and bending of the cytoplasmic end of TM2 lead to a somewhat elevated RMSD of 2.1 Å for 366 Cα atoms from both protomers after alignment based on TM3/6. For D70A_asym, the TMD in the IF conformation has a similar RMSD of 2.6 Å relative to WT, holo (183 Cα atoms from protomer B, *Figure 2D*), whereas the occluded TMD has an elevated RMSD of 5.8 Å (197 Cα atoms from protomer A after aligning TM3/6) due to large differences in the cytoplasmic ends of TM1,2,4,5.

The observed structural effects of the D70A mutation are broadly consistent with MD simulations based on the WT structure. In particular, when site B was empty, the TM2/TM3 loop exhibited greatly increased mobility documented by more than twofold increase in root mean square fluctuations (RMSF, *Figure 3A*); this result is consistent with disorder of this loop in D70A cryo-EM structures. The CTD also exhibited increased mobility as documented both by the broader distribution of the angles relative to the TMD (*Figure 3D*) and by per residue RMSF, in particular when the TMD was used as reference for alignment (*Figure 3B*), indicating that the CTD moved as a fairly rigid body relative to the TMD. In structures with $Zn^{2+}$ bound at site B, the TM2/TM3 loop from one protomer is close to the linker between TM6 and the CTD of the opposing protomer (*Figure 3E and F*, *Figure 1—figure supplement 1B and E*, and N). We therefore used the distance between Cα atoms of Asp72 in the TM2/TM3 loop and Arg210 in the TM6/CTD linker from the opposing chain as a collective variable for tracking CTD movement. The corresponding distance distributions (*Figure 3C*) are broader and extend to larger values when site B is empty. In simulations of the D72A mutant with $Zn^{2+}$ bound at site B there is a marked increase in CTD/TMD angle (*Figure 3D*) as well as a more modest increase in the distance distribution (*Figure 3C*). These results, together with structures of D70A_sym and WT, apo, suggest that a salt bridge between Asp72-Arg210 helps maintain the symmetry seen in the holo

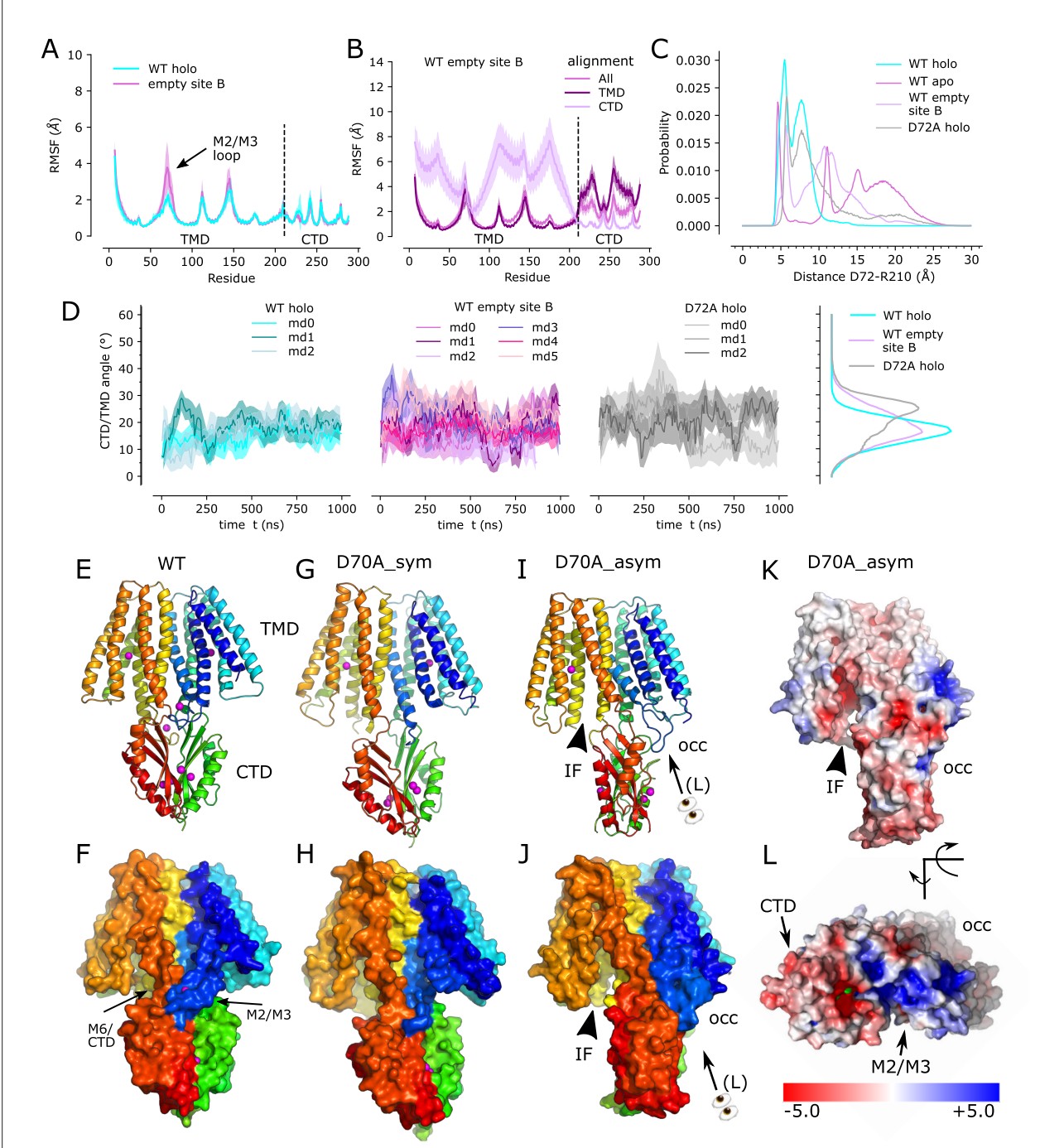

**Figure 3.** Zn$^{2+}$ removal from site B in the TM2/TM3 loop. (**A**) Per-residue RMSF of the WT, holo structure with Zn$^{2+}$ present (cyan) and absent (purple) at site B demonstrates a notable increase in fluctuations in the TM2/TM3 loop. The dashed line indicates the boundary between TMD and CTD. (**B**) Per-residue RMSF for simulations with empty site B using three different alignment schemes: the entire molecule ("All"), transmembrane domain ("TMD"), or C-terminal domain ("CTD"). Analogous data for the WT, holo structure have previously been published (Figure 3i in *Lopez-Redondo et al., 2021*). (**C**) Distributions of distance between Cα atoms from Asp72 in one chain and Arg210 in the opposite chain during simulations of the WT, holo structure, WT apo structure with Zn$^{2+}$ absent from all sites, WT structure with site B empty, and the D72A mutant with Zn$^{2+}$ present at all three sites. The sharp peak at ~5 Å from the holo structure suggests a salt bridge that is less stable in the D72A mutant and disrupted when Zn$^{2+}$ is absent from site B. (**D**) Angle between the TMD and CTD in simulations in the presence (cyan) and absence (purple) of Zn$^{2+}$ at site B; the D72A mutant in the holo state is also shown on the right. The distribution of angles, on the far right, highlight greater mobility either when site B is empty or with the D72A mutation. (**E,F**) Structure of the WT, holo YiiP dimer showing global C2 symmetry about a vertical axis and juxtaposition of the TM2/3 loop (blue) with the TM6/CTD linker from the opposing protomer (orange-to-red). Rainbow colors progress from blue to red moving from the N-terminus of one protomer to the C-terminus of

*Figure 3 continued on next page*

Figure 3 continued

the other protomer. (**G,H**) D70A_sym structure shows a kink between TMD and CTD and disordering of the TM2/3 loop. Both protomers are in the IF state. (**I,J**) D70A_asym structure showing further twisting of the CTD and asymmetry of the TMD's. The TM2/3 loop is disordered in the protomer on the left (chain B), but adopts a novel interaction with the CTD in the protomer on the right (chain A). (**K,L**). Electrostatic surface of D70A_asym showing a negatively charged cavity leading to site A on the left, but an occluded cavity with positive charge on the right. Note that L is at an oblique angle looking down on the M2-M3 loop. Profiles in panels A, B and D represent the average over both protomers and all three simulations, with the mean shown as a solid line and the error band indicating standard deviation.

The online version of this article includes the following source data for figure 3:

**Source data 1.** Source data for *Figure 3* panel A.

**Source data 2.** Source data for *Figure 3* panel B.

**Source data 3.** Source data for *Figure 3* panel C.

**Source data 4.** Source data for *Figure 3* panel D.

state. Release of $Zn^{2+}$ from site B causes disordering of the TM2/TM3 loops leading to loss of the salt bridge and thus greater mobility of the CTD. However, the D70A_asym structure indicates that this disordering is transient, as the loop adopts the extended conformation and associates with different elements of the CTD (*Figure 2C*). Interestingly, this novel change in the TM2/TM3 loop is coupled with movements of TM1,2,4,5 that close site A off from the cytoplasm (*Figure 3I–J*, *Figure 2—figure supplement 2*). Mapping of electrostatic charge shows that whereas the access channel to site A in the IF protomer is negatively charged, positive charge dominates the cytoplasmic surface of the protomer in the occluded conformation (*Figure 3K and L*).

## Site C: D287A and D287A/H263A mutations

The D287A mutant was chosen to disrupt site C since it coordinates both $Zn^{2+}$ ions at this site. However, the corresponding structure (3.7 Å resolution, *Figure 1D*, *Figure 1—figure supplement 4*) shows no structural changes relative to the WT structure (1.18 Å RMSD for all 562 Cα atoms) and $Zn^{2+}$-related densities are still clearly visible at site C (*Figure 1—figure supplement 1O*). We therefore introduced a second mutation, D287A/H263A, and the corresponding particles had an unusual appearance consistent with formation of higher order oligomers (*Figure 1—figure supplement 4*). This observation suggested that site C may be responsible for the integrity of the homodimer. Indeed, SEC elution profiles of the YiiP-Fab complex show that the main peak from D287A/H263A is shifted relative to the other mutants (from 11.1 to 10.1 ml, *Figure 1—figure supplement 2A and B*, 3 A, 4 A,B). Elution profiles from complexes formed with D70A and D287A also have an earlier peak at ~10 ml and, in both cases, image processing revealed a relatively small subpopulation of particles forming a dimer of dimers (*Figure 1—figure supplement 3C* & 4E). However, for D287A/H263A, this dimer of dimers represents the main peak and the only class of particles that could be refined to high resolution (3.5 Å; *Figure 1—figure supplement 4H*). For D287A/H263A, higher order oligomers are apparent during 2D and 3D classification (*Figure 1—figure supplement 4D and F*) and are likely to explain the earlier elution peak at 8–9 ml.

The dimer of dimers from D287A/H263A is stabilized by an unprecedented domain swap in which CTD's and TMD's affiliate with different protomers (*Figure 2G*). This domain swap is enabled by a structural transition of the linker between the TMD and the CTD. In particular, the normally unstructured loop between TM6 and the first helix of the CTD (H1) is reconfigured into a very long, continuous α-helix extending from the N-terminus of M6 (Trp178) to the C-terminus of H1 (Glu226). Site C itself is completely disordered and the CTD's are splayed apart (*Figure 2G*, *Figure 2—figure supplement 1C*): that is, the distance between Cα atoms of Arg237 and Glu281 is 22.7 Å compared to ~12 Å in structures with an intact site C (*Lopez-Redondo et al., 2021*). This domain swap was not observed in the dimer-of-dimers formed by D70A and D287A, where the CTD and site C were unperturbed and Fab molecules were uniquely responsible for inter-dimer interactions. Although it was not possible to refine a structure from the higher order oligomers from D287A/H263A, due to heterogeneity and preferred orientation, they appeared to comprise a linear chain of molecules propagated via this domain swap. These surprising changes in the CTD indicate that site C is indeed crucial in maintaining the integrity of the native homo-dimer. Despite the domain swap, however, TMD's from the D287A/H263A complex are quite congruent with the WT structure (RMSD 1.35 Å for 346 Cα atoms excluding

the TM2/TM3 loop) and density for $Zn^{2+}$ ions is clearly visible at site A (*Figure 1—figure supplement 1P*). Weak density is visible for the TM2/TM3 loops, indicating that they adopt an extended configuration that disrupts site B and allows them to interact with the CTD; it does not appear that $Zn^{2+}$ is bound at site B in this domain-swapped complex (*Figure 1—figure supplement 1Q* and *Figure 2—figure supplement 1C and D*).

## $Zn^{2+}$ binding affinity

To assess the binding affinity of individual $Zn^{2+}$ binding sites, we used MST to analyze a series of mutants designed to isolate the individual sites (*Figure 4* and *Figure 4—figure supplement 1*). Specifically, the triple mutant D70A/D287A/H263A was used to study site A, D51A/D287A/H263A to study site B, and D51A/D70A to study site C. To maintain accurate and reproducible $Zn^{2+}$ concentrations ranging from nanomolar to micromolar, we used nitrilotriacetic acid (NTA, $K_d$ = 14 nM) to buffer $Zn^{2+}$ for sites with high affinity, and citrate ($K_d$ = 12 µM) for sites with lower affinity. At pH 7, sites A and C displayed relatively high affinity ($K_d$ = 16 and 33 nM, respectively, *Table 2*, *Figure 4*), whereas site B had considerably lower affinity ($K_d$ = 1.2 µM). Site C is binuclear, and the apparent affinity was reduced when further mutations were introduced to isolate individual C1 (D51A/D70A/H263A with $K_d$ = 153 nM) and C2 (D51A/D70A/H234A with $K_d$ = 223 nM) sites, suggesting cooperative binding of ions at C1 and C2 as implied by the coordination geometry of this site (*Figure 1D* inset).

To explore the basis for coupling of $Zn^{2+}$ transport to the proton-motive force, we measured $Zn^{2+}$ binding affinity at pH values from 5.6 to 7.4. We found that affinity at site A changed by five orders of magnitude: $K_d$ ranging from 1 nM at pH 7.4 - consistent with the cytosol - to 302 µM at pH 5.6 (*Figure 4A*, *Table 2*). Site B had only modest pH dependence ranging from 1 to 16.6 µM, whereas Site C changed by two orders of magnitude from 0.033 to 6.7 µM. Cooperativity at site C is consistent with a high Hill coefficient at pH 7 (n=2.9), which fell below 1 at lower pH's. However, Hill coefficients obtained for other sites were quite variable (*Table 3*), including those for sites C1 and C2, making conclusions about cooperativity inconclusive.

The domain swap seen in cryo-EM structures of D287A/H263A raises concern that site C mutations might affect affinity measured at other sites. However, in the absence of Fab, all of the mutants used for MST studies eluted from SEC columns at the same volume as WT (*Figure 4—figure supplement 1A-D*) indicating that they all adopted the native dimeric structure. This observation suggests that the domain-swapping seen in the structure of D287A/H263A was induced by the Fab molecules. This conclusion is supported by direct comparison of preparations with site C mutations before and after addition of Fab (*Figure 4—figure supplement 1I and J*). In the absence of Fab, they both elute from SEC as a single peak at the expected volume of 12 ml. In the presence of Fab, there is a shift of the main peak consistent with formation of a higher molecular weight complex and, additionally, appearance of a second peak. In the case of D287A-Fab, the main peak at 11.1 ml is consistent with the YiiP-Fab complex seen for the other mutants (*Figure 1—figure supplements 2 and 3*) and the second peak at 9.9 ml is consistent with the dimer of native homodimers seen during image processing. In the case of D287A/H263A, the main peak in the presence of Fab is at 10.1 ml and thus consistent with a dimer of domain-swapped dimers and the secondary peak at 8.6 ml is consistent with higher order oligomers seen during cryo-EM processing (*Figure 1—figure supplement 4*). In addition, we performed MST analysis with the D70A/D287A mutation as an alternative for measuring affinity at site A. The cryo-EM structure of D287A shows that the native homodimeric assembly is retained and although $Zn^{2+}$ was observed at site C, we expect it will bind at lower affinity and thus not overlap with higher affinity binding at site A. Indeed, this construct produced Kd=~1 nM at pH 7 and ~6 µM at pH 6, which is consistent with results from D70A/D287A/H263A (*Table 2*), thus confirming the pH dependence of site A.

We also used MD simulations in conjunction with the experimental MST data to address pH dependence of $Zn^{2+}$ binding and to evaluate the contribution of individual residues. To start, we estimated pKa values for each titratable residue at site A (Asp47, Asp51, His155 and Asp159) and site B (Asp70, His73 and His 77) using constant pH MD (CpHMD) simulations with pH replica exchange (*Huang et al., 2021*) in the absence of $Zn^{2+}$. It was not possible obtain data for site C because the $Zn^{2+}$-free CTD proved unstable in CpHMD simulations. These simulations generated probabilities for each protonation state as a function of pH; these probabilities were fit with the Hill-Langmuir equation to derive per-residue $pK_a$'s (*Figure 4—figure supplements 2–4*, *Table 4*). For site A residues, His155

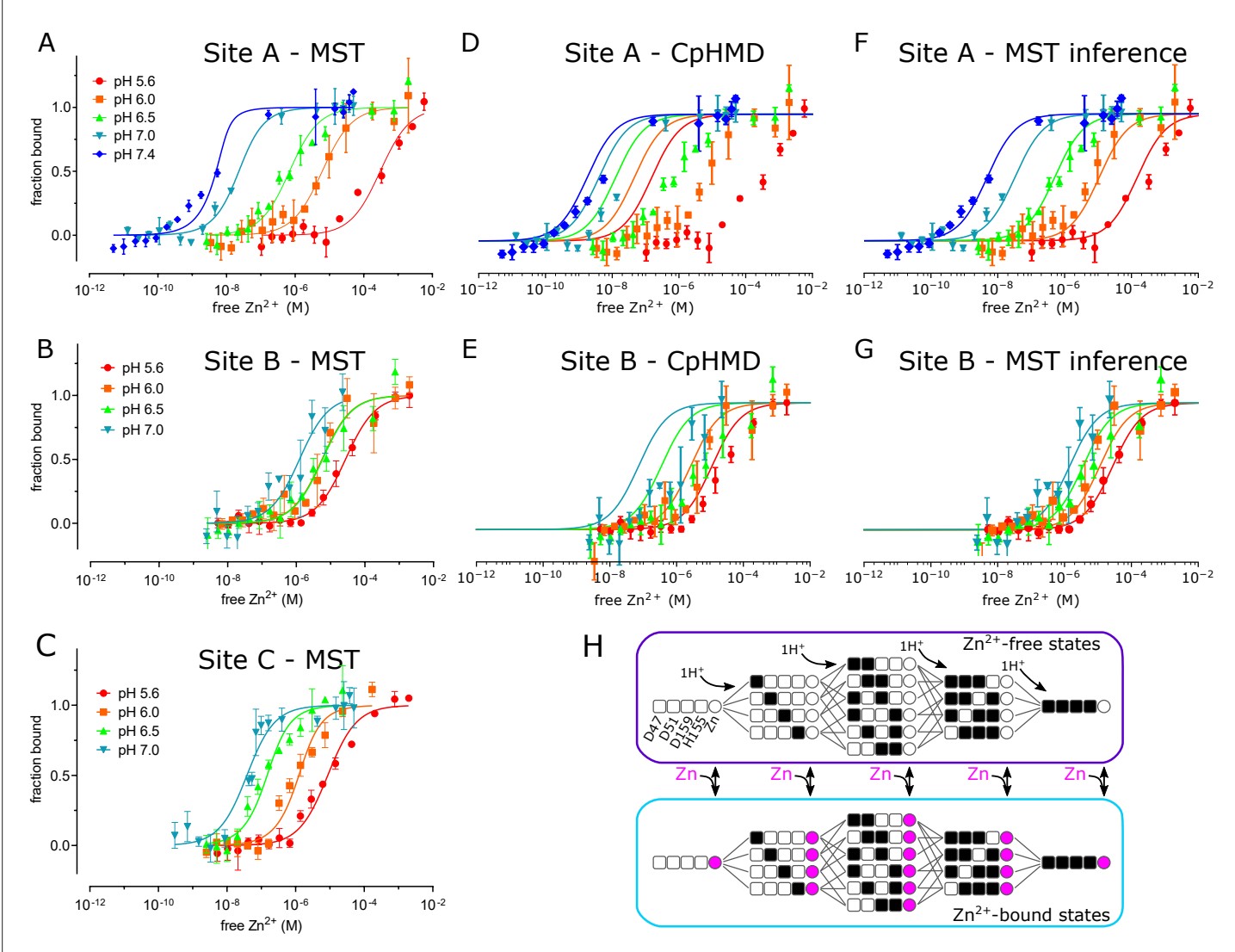

**Figure 4.** pH dependence of Zn²⁺ binding. MST was used to experimentally measure Zn²⁺ binding affinity at each site (panels A, **B and C**). These data were combined with a thermodynamic model, represented in panel H, with parameters derived either from CpHMD (panels D and E) or MST inference (panels **F and G**). (**A**) For MST studies of site A, the D70A/D287A/H263A construct was used with Zn²⁺ buffered either with NTA (pH 7.0 and 7.4) or with citrate (pH 5.6, 6.0, and 6.5). Curves represent the law of mass action with $K_d$ values listed in *Table 2* together with their 95% confidence intervals. The relatively poor fit at pH 7.4 may reflect the fact that the $K_d$ (1 nM) is lower than the minimum protein concentration (8 nM) supported by the assay, thus moving the system from the binding to the titration regime (*Jarmoskaite et al., 2020*). (**B**). For MST studies of site B, the D51A/D287A/H263A construct was used and Zn²⁺ was buffered with citrate. (**C**) For MST studies of site C, the D51A/D70A construct was used with NTA at pH 7 or with citrate at the other pH's. (**D,E**) Curves represent predictions of a thermodynamic model with p$K_a$ values for either site A residues (Asp47, Asp51, His153, Asp157) or site B residues (D70, H73, H77) taken from CpHMD simulations. These p$K_a$ values are listed in *Table 4* and the distributions of protonation states are shown in *Figure 4—figure supplements 5 and 6*. Symbols represent experimental MST data as in A and illustrate a poor fit using these parameters. Thermodynamic modeling was not possible for site C because of instability of the CTD in CpHMD simulations in the absence of Zn²⁺. (**F,G**). Curves represent predictions of the MST inference algorithm with corresponding p$K_a$ values listed in *Table 4*. Again, symbols represent experimental MST data and show the excellent fit using these refined parameters. (**H**) Schematic representation of the microscopic thermodynamic model for site A. Titratable residues (D47, D51, D159, H155) are represented either as black squares for the protonated state, or white squares for the deprotonated state. Binding of one Zn²⁺ ion to the site is indicated by a filled magenta circle. Transitions are only possible between states connected by lines (protonation/deprotonation) or corresponding states connected by double-sided arrows (Zn²⁺ binding/release). An analogous model for site B comprised D70, H73 and H77 (not depicted). MST data was collected in triplicate from three independent titrations. Protein used for these titrations represent a single biological replicate, although qualitatively similar results were obtained from three to five independent protein preparations derived from at least two bacterial cultures.

The online version of this article includes the following source data and figure supplement(s) for figure 4:

*Figure 4 continued on next page*

*Figure 4 continued*

**Source data 1.** Source data and fitting for the plots in of site A in panel A.

**Source data 2.** Source data and fitting for the plots in of site B in panel B.

**Source data 3.** Source data and fitting for the plots in of site C in panel C.

**Figure supplement 1.** Measuring $Zn^{2+}$ affinity by MST.

**Figure supplement 1—source data 1.** Source data plotted in panels A-D.

**Figure supplement 1—source data 2.** Source data for the MST titrations of Site A with Zn shown in panel E.

**Figure supplement 1—source data 3.** Source data for the MST titrations of Site B with Zn shown in panel F.

**Figure supplement 1—source data 4.** Source data for the MST titrations of Site C with Zn shown in panel G.

**Figure supplement 1—source data 5.** Source data for the MST titrations of Site A with Mg shown in panel H.

**Figure supplement 1—source data 6.** Source data plotted in panels I-J.

**Figure supplement 2.** Walk of replica simulations in CpHMD pH-replica exchange (REX) through pH space.

**Figure supplement 2—source data 1.** Source data documenting the state transitions of each replica simulation.

**Figure supplement 3.** Convergence of the deprotonated fraction for titratable residues in CpHMD simulations.

**Figure supplement 3—source data 1.** Source data for changes of the proton-bound fractions of specified residues at different pH over the course of the simulation.

**Figure supplement 4.** Titration curves for titratable residues in CpHMD simulations.

**Figure supplement 4—source data 1.** Source data for titration curves of the residues in binding sites A and B.

**Figure supplement 5.** Site A Protonation and $Zn^{2+}$-binding states by CpHMD and MST inference.

**Figure supplement 5—source data 1.** Source data for values plotted in panel A.

**Figure supplement 5—source data 2.** Source data listing pKa values deduced from CpHMD simulations.

**Figure supplement 5—source data 3.** Source data listing pKa and Kd values deduced from MST-inference method.

**Figure supplement 6.** Site B Protonation and $Zn^{2+}$-binding states by CpHMD and MST inference.

**Figure supplement 6—source data 1.** Source data for values plotted in panel A.

**Figure supplement 6—source data 2.** Source data listing pKa values deduced from CpHMD simulations.

**Figure supplement 6—source data 3.** Source data listing pKa and Kd values deduced from MST-inference method.

---

had the highest predicted $pK_a$ (8.0), followed by Asp159 (4.9), Asp47 (3.8) and Asp51 (2.9). Predicted populations of the individual microstates (s0-s15, *Figure 4—figure supplement 5A and B*) indicated that two residues are likely protonated at pH 5.5 with His155 getting protonated first followed by Asp159. Although these simulations were conducted with the IF conformation, we do not expect significant differences for the OF state, given the similarity of site A seen in the X-ray structure (*Lu et al., 2009*) of the OF state (RMSD of 0.49 Å for 17 Cα atoms on TM2 and TM5 and 1.01 Å for all atoms composing the four residues that coordinate the $Zn^{2+}$ ion). For site B, His73 and His77 have

**Table 2.** Binding affinity for individual $Zn^{2+}$ sites measured by MST* or deduced by the MST inference algorithm[†].

| | site A D70A/ D287A/ H263A (µM) | site A/C D70A/ D287A (µM) | site B D51A/ D287A/ H263A (µM) | site C D51A/D70A (µM) | site C1 D51A/ D70A/ H263A (µM) | site C2 D51A/ D70A/ H234A (µM) |
|---|---|---|---|---|---|---|
| pH 5.6 | 302±107 (149) | | 27.1±2.8 (19.5) | 8.10±1.55 | | |
| pH 6.0 | 6.11±1.32 (10.8) | 6.30±1.94 | 5.54±2.0 (5.38) | 0.692±0.28 | | |
| pH 6.5 | 0.654±0.173 (0.503) | | 5.73±1.75 (1.36) | 0.088±0.032 | | |
| pH 7.0 | 0.0163±0.0041 (0.0325) | 0.0012±0.0018 | 1.18±0.6 (0.503) | 0.033±0.0087 | 0.153±0.048 | 0.223±0.039 |
| pH 7.4 | 0.001±0.001 (0.0048) | | | | | |

*Values correspond to $K_d$ as determined by applying the law of mass action to the data together with their 68% confidence intervals.

[†]Values deduced by MST inference are shown in parentheses.

---

**Table 3.** Binding affinity and Hill coefficients derived from MST data*.

| | site A D70A/ D287A/H263A (µM) | site B D51A / D287A/H263A (µM) | site C D51A/D70A (µM) | site C1 D51A/ D70A/H263A (µM) | site C2 D51A / D70A/H234A (µM) |
|---|---|---|---|---|---|
| pH 5.6 | 1490±2,460 n=0.46 | 28.4±3.54 n=0.94 | 7.50±2.20 n=0.9 | | |
| pH 6.0 | 6.34±1.67 n=0.81 | 5.68±1.57 n=1.60 | 1.39±0.382 n=0.78 | | |
| pH 6.5 | 0.622±0.198 n=0.54 | 27.9±34.3 n=0.38 | 0.133±0.0359 n=0.66 | | |
| pH 7.0 | 0.0247±0.0057 n=1.0 | 4.23±13.0 n=0.45 | 0.0486±0.0038 n=2.9 | 0.116±0.0119 n=2.96 | 0.212±0.0328 n=1.2 |
| pH 7.4 | 0.0039±0.0009 n=0.56 | | | | |

*Values correspond to $EC_{50}$ and the Hill coefficient (n) as determined by fitting the Hill equation to the data together with their 68% confidence intervals.

almost identical $pK_a$'s of 8.1, and the dominant state at neutral pH consists of both His73 and His77 protonated (**Figure 4—figure supplement 6A and B**).

We went on to use a thermodynamic model incorporating both protonated and $Zn^{2+}$-bound states to assess the stoichiometry of transport (**Figure 4H**). We used this model to generate $Zn^{2+}$ binding curves based on $pK_a$'s of individual residues and binding free energies for $Zn^2$ derived from the experimental MST data (**Table 2**). Initially, $pK_a$'s of each residue in the absence of $Zn^{2+}$ were taken from the Hill-Langmuir fits to the CpHMD data (**Table 4**, **Figure 4—figure supplement 4**). However, this CpHMD-based model did not agree with the experimental data (**Figure 4D and E**), likely due to inaccuracy in microstate $pK_a$ values. We therefore employed a novel modeling method based on the inverse *Multibind* approach (see Materials and Methods) to iteratively refine the CpHMD $pK_a$'s and the MST $Zn^{2+}$ binding free energies using the MST data points as a target for Monte Carlo (MC) minimization. This analysis, referred to henceforth as *MST inference*, allowed us to estimate probabilities of all the microstates as a function of pH and $Zn^{2+}$ concentration (**Figure 4—figure supplements 5 and 6**). After refinement of parameters by MST inference, the predicted $Zn^{2+}$ binding curves closely reproduced the observed pH-dependence of $Zn^{2+}$ binding at both sites A and B (**Figure 4F and G**). The

**Table 4.** $pK_a$ values of residues determined by CpHMD simulations (Hill-Langmuir) and MST inference.

| | | CpHMD (Hill-Langmuir)* | MST inference† |
|---|---|---|---|
| Site A | Asp47 | 3.84±0.18 | 0.20±0.51 |
| | Asp51 | 2.92±0.12 | 6.51±0.21 |
| | His155 | 7.97±0.15 | 7.82±0.15 |
| | Asp159 | 4.87±0.18 | 7.83±0.15 |
| Site B | Asp70 | 2.06±0.46 | 1.12±0.57 |
| | His73 | 8.08±0.46 | 12.37±0.61‡ |
| | His77 | 8.12±0.13 | 5.26±0.47‡ |

*For CpHMD(Hill-Langmuir), CpHMD data for protomer A and B were collected and analyzed independently. The value is the mean over the protomer A and B data and the error estimate is the average of the absolute deviations from the mean.

†For MST inference, error estimates were obtained from 50 independent replicates of the MC sampling process. The value is the mean over the 50 replicates and shown with the standard deviation.

‡Titration curves for His73 and His77 from the MST inference model cannot be fit individually with a simple Hill equation (c.f. Supp. Figs. 11e & f). We therefore consider these two residues as a coupled system with two effective $pK_a$ values as shown.

dramatically improved agreement is a consequence of substantial changes in the p$K_a$'s of individual residues based on the MST inference algorithm (**Table 4**).

A more detailed look at the population of microstates predicted by MST inference allows us to explore the interplay of Zn$^{2+}$ and H$^+$ binding under physiological conditions. For site A in the absence of Zn$^{2+}$, the dominant state at cytosolic pH of 7.5 consists of 2 protons bound by His155 and Asp159 (**Figure 4—figure supplement 5C and D**). Based on the microscopic p$K_a$ values, we calculated the 'coupling energy' (**Ullmann, 2003**) and thus deduced that these two residues form a coupled system (see Materials and Methods). As a result, protonation of His155 and Asp159 is highly cooperative and produces a very steep binding curve (**Figure 4—figure supplement 5E**); the singly protonated state is strongly suppressed and the effective p$K_a$ is 7.8 for both residues (**Figure 4—figure supplement 5D**). At the lower pH of the periplasm, a third proton is recruited by Asp51, whose p$K_a$ increased to 6.5 during refinement by MST inference (**Table 4**). Regardless of the pH, as Zn$^{2+}$ binds at site A, all the residues are deprotonated (**Figure 4—figure supplement 5F**), supporting the idea that Zn$^{2+}$ transport can be coupled with a proton gradient across the plasma membrane. For site B, the dominant state at cytosolic pH in the absence of Zn$^{2+}$ has one protonated residue: either His73 or His77 (**Figure 4—figure supplement 6c and D**). Although the microstate p$K_a$'s are equivalent, these site B residues behave quite differently from Site A residues and generate an anti-cooperative binding curve with effective p$K_a$'s of 5.6 and 12.4 (**Figure 4—figure supplement 6e and F**). Zn$^{2+}$ binding at cytosolic pH displaces all protons (**Figure 4—figure supplement 6G**). Even though protons may co-exist with Zn$^{2+}$ at lower pH's, this is not a physiologically relevant condition given that site B is uniquely exposed to the cytoplasm.

## Discussion

In this study, we have focused on the properties of individual Zn$^{2+}$ binding sites of YiiP from *S. oneidensis*. Previous studies have shown that this protein forms a homodimer in the IF state with Zn$^{2+}$ ions constitutively bound at three sites (**Lopez-Redondo et al., 2018**). Treatment with EDTA to remove Zn$^{2+}$ from all three sites induced a conformational change to produce an occluded state, a necessary precursor to the OF state (**Lopez-Redondo et al., 2021**). For the current work, we generated mutations at each site to measure their binding affinities and to assess their respective roles in generating conformational change. We found that site A, considered to be the transport site in the middle of the TMD, has nanomolar affinity at a cytoplasmic pH of 7.4, which is reduced 10$^5$-fold at the more acidic pH of 5.6. Release of Zn$^{2+}$ from site A induces only modest movement of the peripheral transmembrane helices, but has no global effect on the architecture of the homodimer. Site B, on the short loop between TM2 and TM3, has much lower affinity in the micromolar range with relatively little pH dependence. When site B lacks Zn$^{2+}$, the loop becomes disordered and the protein undergoes a global conformational change leading to an occluded conformation in one of the protomers. Site C, a binuclear site in the CTD that engages residues from both protomers, has relatively high affinity and intermediate pH dependence. When site C was fully disrupted with a double mutation, the homodimer became destabilized such that the Fab molecules used for cryo-EM imaging induced a domain swap of the CTD leading to higher order oligomers.

### Elements stabilizing the homodimer

Dimerization appears to be a universal feature of Cation Diffusion Facilitators. All structures so far show similar elements contributing to a common dimer interface (**Figure 5—figure supplement 1**). In particular, a conserved salt bridge exists at the cytoplasmic surface of the membrane (Lys79-Asp209 for soYiiP); the cytoplasmic end of the CTD generally includes one or more Zn$^{2+}$ sites, and extensive hydrophobic interactions involving TM3 helices mediate TMD interactions (**Lopez-Redondo et al., 2021**). In early work, the salt bridge was proposed to act as a fulcrum for alternating, scissor-like movements in TMD and CTD driven by relay of Zn$^{2+}$ between sites A and C (**Lu et al., 2009**). Furthermore, the original X-ray structure of ecYiiP featured a V-shaped architecture in which TMD's were completely disengaged (**Lu et al., 2009**; **Lu and Fu, 2007**). However, subsequent work showed that stabilization of TMD interactions by cysteine crosslinking did not inhibit transport activity (**Lopez-Redondo et al., 2018**) and compact dimeric TMD interfaces were observed in cryo-EM structures of soYiiP, Znt8, and Znt7 (**Bui et al., 2023**; **Lopez-Redondo et al., 2018**; **Xue et al., 2020**). Indeed, the TMD interface is

a remarkably conserved feature in our new structures of D70A and D287A/H263A mutants, despite large-scale conformational changes involving the TM2/TM3 loop and domain swapping of the CTD. We therefore conclude it to be unlikely that the TMD's undergo large-scale scissoring movements as part of the transport cycle.

For isolated CTD's, metal ion dependent scissor-like movements have been well documented using truncated constructs from a variety of species (*Cherezov et al., 2008*; *Udagedara et al., 2020*; *Zeytuni et al., 2014*). We observed similar movements in full-length soYiiP: i.e., the CTD's move apart when $Zn^{2+}$ is removed from site C either by chelation or mutation. However, other elements must contribute, given that isolated CTD's dimerize even in the absence of metal ions and $Zn^{2+}$ ions were not observed at all in the CTD of Znt7 (*Bui et al., 2023*). Nevertheless, the Fab-induced domain swap seen in D287A/H263A indicates that, for soYiiP, CTD interactions are weakened in the absence of $Zn^{2+}$. Conversely, it seems likely that dimer stabilization, via TMD and salt-bridge interactions, lead to enhanced $Zn^{2+}$ binding in the CTD, as evidenced by the considerably higher $Zn^{2+}$ affinity at site C of full-length soYiiP ($K_d$ of 33 nM at pH 7) compared to those measured from isolated CTD's ($K_d$ in micromolar range *Udagedara et al., 2020*; *Zeytuni et al., 2014*). In any case, the relatively high affinity of site C at pH 7 suggests that this site will remain occupied and will serve to ensure dimer stability under physiological conditions.

## Role of the site B

Structural effects of the D70A mutation support our previous hypothesis that site B is responsible for inducing a global conformational change. The most conspicuous change is the kink between TMD and CTD, which breaks the global twofold symmetry of structures that retain $Zn^{2+}$ binding at site B, in which local twofold axes of CTD and TMD are aligned (*Figure 5—figure supplement 1*). Although stable, kinked conformations were not achieved over the time-scale of MD simulations, increased movement of the CTD relative to the TMD is seen both in the previously reported apo state simulations (*Lopez-Redondo et al., 2021*) and with an empty site B (*Figure 3*), indicating a role of the $Zn^{2+}$-bound TM2/TM3 loop in stabilizing the position of the CTD. Indeed, van der Waals interactions are observed between the structured loop from one protomer and the TM6-CTD linker of the opposing protomer on both sides of the dimer (*Figure 3E and F*), and MD simulations indicate that a salt bridge between Asp72 and Arg210 may reinforce these interactions (*Figure 3D*; see also site B in *Figure 1—figure supplement 1*).

Both cryo-EM structures and MD simulations show that the TM2/TM3 loop becomes disordered when $Zn^{2+}$ is released from site B and this is a likely explanation for increased mobility of the CTD. Two structures were obtained from the D70A mutation in which the CTD becomes progressively more tilted and twisted. In one structure, the TMD's are symmetrical and retain the IF conformation (D70A_sym), whereas in the other structure with more extreme CTD movement (D70A_asym), the TMD transitions to an occluded state. We speculate that these structures represent a sequence of conformational change leading to the occluded state. Although both TM2/TM3 loops are disordered in the D70A_sym structure, one of these loops reforms into an extended conformation in the occluded protomer of the D70A_asym structure (*Figure 2—figure supplement 1*). This change results in a novel interaction between the loop and the CTD, which may drive the displacement and bending of membrane helices thus leading to occlusion of the transport site A (*Figure 2—figure supplement 2*). This interaction would also prevent the TM2/TM3 loop from rebinding $Zn^{2+}$ until the protein returned to the IF state. On the other side of the dimer, the CTD has lost contact with the TMD, allowing the TM2/TM3 loop to remain disordered and the TMD to remain in the IF state. Interestingly, the TM2/TM3 loops are also extended in the D287A/H263A domain-swapped structure (*Figure 2—figure supplement 1*), but in this case, the interactions of this loop with the reconfigured CTD's are quite different and the TMD's remain in the IF state, indicating that a specific structural constraint is required to instigate occlusion.

Occlusion of the transport site A is a result of bending of TM5 and tilting of TM1 and 4, thus narrowing the gap between these helices and closing off access to the cytosol. These movements also generate a dramatic difference in the electrostatic surface at the cytoplasmic side of the TMD. In the IF state, the open cavity leading toward site A is negatively charged, as also shown for Znt7 (*Bui et al., 2023*), thus serving to attract $Zn^{2+}$ toward this site. After transition to the occluded state, not only is the cavity closed (*Figure 2—figure supplement 2*), but the surface becomes positively charged thus repelling $Zn^{2+}$ (*Figure 3K and L*). Previous work has identified a 'hydrophobic gate' consisting of two

residues at the cytoplasmic ends of TM5 and TM6: Leu154 and Leu199 in soYiiP (*Gupta et al., 2014*). Our previous comparison of holo and apo states showed that these two residues do come closer together in the occluded state (*Lopez-Redondo et al., 2021*) and homologous residues in Znt7 are close in the OF state and separated in the IF state (*Bui et al., 2023*). Interaction of Leu154 and Leu199 is also seen in the occluded protomer in the D70A_asym structure (*Figure 2D*), but these residues are at the periphery of the cavity leading to site A. This observation suggests that interaction of these residues may play an important role in stabilizing the occluded state, but that bending at the cytoplasmic end of TM5 (Val148-Ala151) and tilting of TM1 and TM4 may be more directly responsible for blocking access to the transport site.

For soYiiP, the IF state appears to be a low-energy, ground state when $Zn^{2+}$ is present at all three sites. Indeed, the IF state has been seen for WT protein both in lipid-based helical crystals (*Lopez-Redondo et al., 2018*) and detergent micelles (*Lopez-Redondo et al., 2021*), as well as in D287A and D287A/H263A mutant structures, all with RMSD's<1.5 Å. Removal of $Zn^{2+}$ from site B can be viewed as a source of energy for inducing conformational change and likely leading to transport. Interestingly,

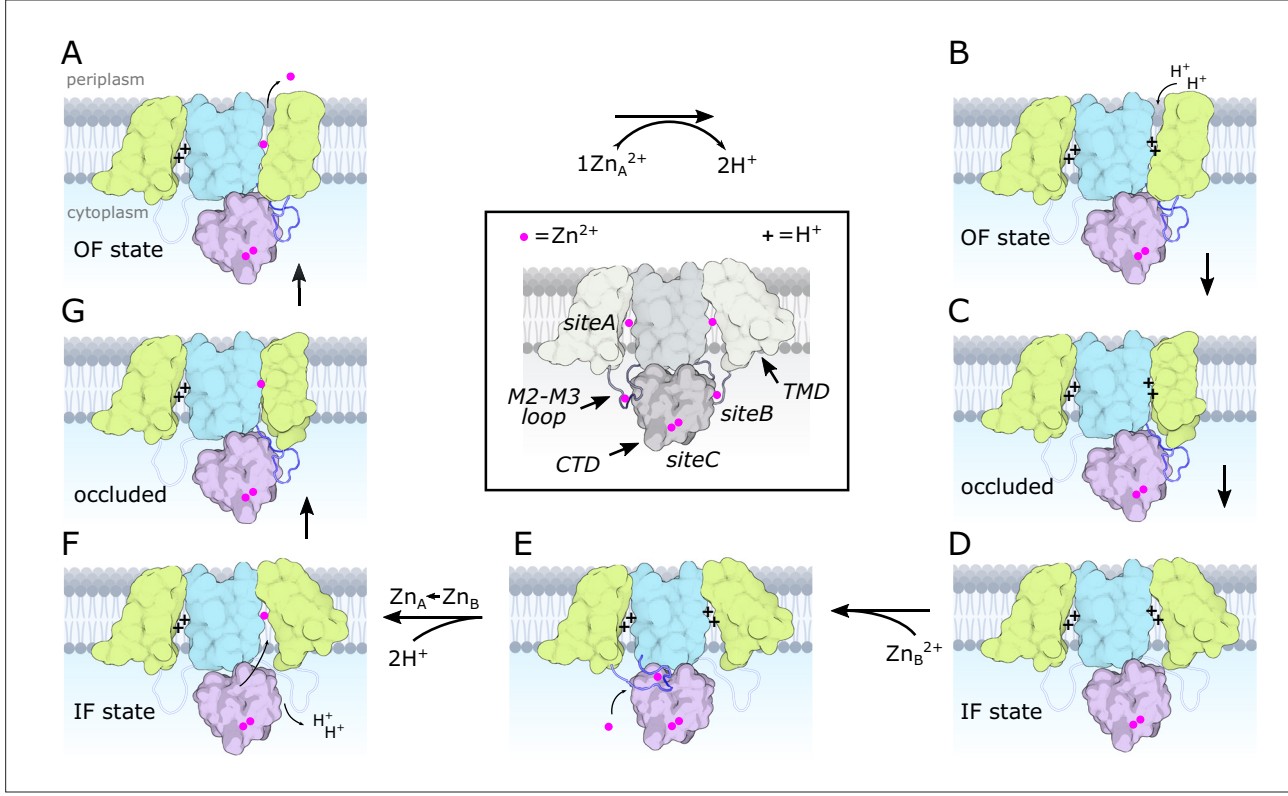

**Figure 5.** Transport cycle for YiiP. According to the alternating access paradigm, YiiP toggles between the OF and IF states via an intermediate occluded state. We assume that these changes occur independently in each protomer based on asymmetry seen in structures of soYiiP and Znt8 (*Xue et al., 2020*). Thus, our model depicts changes in the right-hand protomer while the left-hand protomer remains in a resting IF state. Although these intermediate states are informed by current and past structural work, they do not precisely conform to solved structures but represent hypothetical states that we believe to exist under physiological conditions. (**A**) $Zn^{2+}$ is released to the periplasm. The TM2/TM3 loop is depicted interacting with the CTD in a $Zn^{2+}$-free, extended conformation, as seen in our D70A_asym structure. (**B**) The release of $Zn^{2+}$ is promoted by the low pH of the periplasm and results in protonation of two residues in site A, or potentially three residues at lower pH. (**C–D**) The protonated form transitions to the IF state via an occluded state. In the IF state, the $Zn^{2+}$-free TM2/TM3 loop is released by the CTD and becomes disordered. (**E**) $Zn^{2+}$ is recruited to site B, inducing an ordered conformation of the TM2/TM3 loop that folds onto the CTD enabling interaction of Asp72 and Arg210. (**F**) $Zn^{2+}$ is transferred from a relatively low affinity site B to the much higher affinity site A via a negatively charged access channel, thus displacing two protons. (**G**) This transfer induces a $Zn^{2+}$-bound, occluded conformation in which the CTD tilts toward the occluded protomer and interacts with the TM2/TM3 loop in its $Zn^{2+}$-free, extended conformation. Features of the model are illustrated in the middle, boxed panel with desaturated colors. $Zn^{2+}$ ions are depicted as magenta spheres and protons with a '+'. The CTD is pink with two $Zn^{2+}$ ions constitutively bound at site C. The scaffolding membrane helices (TM3 and TM6) are blue and the transport domain (TM1,2,4,5) is yellow. The TM2/TM3 loop is blue and depicted with dashed lines in the disordered state. Created with biorender.com.

The online version of this article includes the following figure supplement(s) for figure 5:

**Figure supplement 1.** Summary of structures from CDF transporters.

the occlusion only occurs in one protomer, suggesting a lack of synchrony across the dimer axis. A mixture of conformational states was also observed for Znt8, in which one protomer adopted the OF state while the other was in the IF state. These observations imply that a given protomer may be able to undergo transport while the other remains inactive, as illustrated in *Figure 5*.

The proposed sequence of conformational change (*Figure 5*) is consistent with the notion that $Zn^{2+}$ is initially recruited to site B and that transfer to site A would act as a trigger for the first step of the transport cycle. Such a two-step mechanism might be necessary if $Zn^{2+}$ were delivered to YiiP as a chelate – for example with metallothionein or glutathione – that would not have direct access to site A. In the case of $Cu^+$, metallochaperones are thought to play a role in delivery (*Robinson and Winge, 2010*), but so far there is very limited evidence for $Zn^{2+}$ metallochaperones (*Chandrangsu et al., 2019*). Nevertheless, essentially all of the $Zn^{2+}$ in the cytoplasm exists in chelated form with vanishingly small concentrations of free $Zn^{2+}$ available for direct binding (*Choi and Koh, 1998*; *Outten and O'Halloran, 2001*). The accessibility of site B makes it a plausible acceptor as an initial recruitment site. We postulate that site B would only be available for binding $Zn^{2+}$ in the IF apo state (*Figure 5D*). Once bound, two features would facilitate transfer to site A: the drastic difference in affinity between sites A and B would generate a large free-energy gradient and the negative charge of the cavity leading to site A, which lies almost directly below site B in the WT, holo structures, would direct the ion toward the transport site. Although the amino acid sequence of site B is not conserved amongst CDF transporters, a $Zn^{2+}$ ion was observed in an analogous juxtamembrane position on Znt8 (*Xue et al., 2020*) and many CDF transporters have histidine-rich loops, typically between TM4 and TM5, which might fulfill a similar role in recruiting $Zn^{2+}$ and orchestrating conformational change. Indeed, for Znt7 this His-rich loop was shown to bind two $Zn^{2+}$ ions leading these authors also to speculate that it plays a role in recruitment and shuttling to the transport site (*Bui et al., 2023*).

## Stoichiometry of transport and energy coupling

The proton-motive force is generally postulated to drive the antiport mechanism utilized by CDF transporters. In the case of YiiP, this antiport involves exchange of $Zn^{2+}$ from the cytoplasm for $H^+$ in the periplasm and the stoichiometry of this process is key to energy coupling. In previous work, the $Zn^{2+}$:$H^+$ stoichiometry has been measured to be 1:1 for ZitB (*Chao and Fu, 2004a*) and deduced by cellular or computational studies to be 1:2 for CzcD (*Guffanti et al., 2002*) and Znt2 (*Golan et al., 2019*), respectively. In addition, isothermal calorimetry was used to deduce a binding stoichiometry for $Cd^{2+}$ of 1:1 for ecYiiP (*Chao and Fu, 2004b*). For the current work, we refined CpHMD simulation data with experimental MST data using the inverse *Multibind* approach (*Kenney and Beckstein, 2023*) to infer the prevalent protonation states for site A and thus to address stoichiometry. Our analysis is consistent with $Zn^{2+}$ binding to a fully unprotonated site A regardless of the pH. After $Zn^{2+}$ release, the transport site becomes either doubly or triply protonated, depending on the pH. Due to strong coupling between His155 and Asp159, the singly protonated state is essentially not seen. Thus, this model is consistent with a stoichiometry of at least 1:2, possibly 1:3 in more acidic environments (pH <6).

Our analysis predicts significant shifts in p$K_a$ for Zn-binding residues. Such shifts reflect strong Coulomb interactions due to both the low dielectric of the membrane and the clustering of several titratable residues at the binding site. Effects of the environment have been studied both by experimental (*Gayen et al., 2016*; *Isom et al., 2010*; *Morrison et al., 2015*) and computational (*Henderson et al., 2020*; *Panahi and Brooks, 2015*) methods, showing p$K_a$ values up to 9 for acidic residues. Coupling between nearby residues has also been shown to shift p$K_a$ values (*Yue et al., 2017*), with estimated p$K_a$ values of 3 and 11 for a di-aspartyl pH sensor in a pH-sensitive calcium channel serving as a dramatic example (*Chang et al., 2014*). Together with the current work, these examples illustrate the importance of tuning the local environment to harness the energy of the proton motive force.

The $Zn^{2+}$:$H^+$ stoichiometry has crucial physiological consequences for the energetics of transport, which are governed by the Nernst equation describing electrochemical potential. The overall transport cycle can be described as

$$mH_o + nZn_i \rightleftharpoons mH_i + nZn_o$$

with the free energy calculated as

$$\Delta G = RT\ln\frac{\left[H_i\right]^m}{\left[H_o\right]^m} + RT\ln\frac{\left[Zn_o\right]^n}{\left[Zn_i\right]^n} + zFV_m$$

where $m$ is the number of protons, $n$ is the number of $Zn^{2+}$ ions, $z$ is the net charge for the reaction and $V_m$ is the membrane potential. Based on this equation, the higher stoichiometries of 1:2 or 1:3 provide the cell with increased leverage over $Zn^{2+}$ transport. In the absence of membrane potential, for example, a 10-fold proton gradient (ΔpH of 1) can produce $Zn^{2+}$ gradients at equilibrium of up to $10^2$ or $10^3$, respectively, instead of simply 10 for a 1:1 stoichiometry (more generally, $10^{m/n}$). Membrane potential represents an important component of the proton motive force that will influence electrogenic transport, where $z$ is non-zero. In particular, a stoichiometry of 1:1 would produce net positive charge transfer out of the cell and thus would require working against this membrane potential. Given a potential of –80 mV, relatively normal in *E. coli* (*Felle et al., 1980*), a 1:1 stoichiometry would produce an unfavorable energy term that would overcome a chemical gradient of one pH unit. In contrast, a 1:2 stoichiometry would be electroneutral and would therefore be unaffected by membrane potential thus generating a $Zn^{2+}$ gradient of $10^2$ as described above. Finally, a stoichiometry of 1:3 would benefit both from the membrane potential as well as from the increased number of protons to theoretically generate a gradient of $10^{4.4}$.

These calculations are based on equilibrium thermodynamics, but the pH dependence of $Zn^{2+}$ binding also has implications for the kinetics of transport. In particular, the dramatic difference in $K_d$ at cytoplasmic pH of 7.4 (1 nM) compared to more acidic pH's in the periplasm (e.g., 6 μM at pH 6) implies that the on-rate of $Zn^{2+}$ is dominant in the cytoplasm but that the off-rate is enhanced in the periplasm, assuming that the binding site in the OF state has similar pH dependence. This assumption seems plausible given the similarity of transport site geometry of YiiP in the OF state (*Lu et al., 2009*). Thus, although the final equilibrium concentrations of $Zn^{2+}$ are not affected by the pH dependence of the transport site, the rate of transport and the rate of equilibration will be greatly enhanced (*Tanford, 1983*).

YiiP has three aspartates and one histidine at site A, whereas many CDF transporters including the mammalian Znt's substitute Asp47 for a second histidine. Previous work has focused on changes in ion specificity associated with this change (*Hoch et al., 2012*), but it also seems likely to affect stoichiometry. At a minimum, two histidines would make the 1:2 stoichiometry almost certain and a 1:3 stoichiometry highly plausible. The resulting electrogenicity could be desirable for eukaryotic cells in which these transporters operate predominantly in intracellular organelles such as insulin secreting granules, synaptosomes, golgi or zinc-o-somes that are responsible for the dramatic $Zn^{2+}$ sparks during oocyte fertilization (*Chu, 2018*; *Hara et al., 2017*). Since the pH gradient of these organelles is modest, additional driving force from the membrane potential might be key in producing the high internal concentrations that are sometimes required for function.

# Materials and methods

## Key resources table

| Reagent type (species) or resource | Designation | Source or reference | Identifiers | Additional information |
|---|---|---|---|---|
| Gene (*Shewanella oneidensis*) | fieF, SO_4475, | This study | Q8E919 | See *Coudray et al., 2013* |
| Strain, strain background (*Escherichia coli*) | BL21-CodonPlus (DE3)-RIPL | Agilent | Part Number: 230280 | |
| Strain, strain background (*Escherichia coli*) | strain 55244 | ATCC | 27C7 | |
| Recombinant DNA reagent | pET-YiiP (plasmid) | This study, *Lopez-Redondo et al., 2021* | | See *Coudray et al., 2013* |

*Continued on next page*

*Continued*

| Reagent type (species) or resource | Designation | Source or reference | Identifiers | Additional information |
|---|---|---|---|---|
| Recombinant DNA reagent | pET_Fab2r (plasmid) | *Lopez-Redondo et al., 2021* | | |
| Sequence-based reagent | D51A_F | This study | PCR primer | GATTCTTTTGCCGCTACGCTCGCCTCG |
| Sequence-based reagent | D51A_R | This study | PCR primer | CGAGGCGAGCGTAGCGGCAAAAGAATC |
| Sequence-based reagent | D70A_F | This study | PCR primer | GTCCCTGCTGCTCATGACCACAGATACGGCC |
| Sequence-based reagent | D70A_R | This study | PCR primer | GTGGTCATGAGCAGCAGGGACAATGGCATAAC |
| Sequence-based reagent | D287A_F | This study | PCR primer | GATTATTCACCAAGCTCCCGTGCAAG |
| Sequence-based reagent | D287A_R | This study | PCR primer | CTTGCACGGGAGCTTGGTGAATAATC |
| Sequence-based reagent | H263A_F | This study | PCR primer | CGAAGCCGCTAGCATTACCGATACAACAGGGC |
| Sequence-based reagent | H263A_R | This study | PCR primer | CGAAGCCGCTAGCATTACCGATACAACAGGGC |
| Peptide, recombinant protein | Fab2R | This study, *Lopez-Redondo et al., 2021* | | See Methods and materials |
| Peptide, recombinant protein | Tobacco Etch Virus (TEV) protease. | This study | | Previously synthesized recombinantly in *E. coli* cells. |
| Chemical compound, drug | N,N,N',N'-tetrakis(2-pyridinylmethyl)–1,2-ethanediamine [TPEN] | Sigma-Aldrich | CAS number: 16858-02-9 | |
| Chemical compound, drug | Alexa Fluor 488 NHS Ester (Succinimidyl Ester) | Invitrogen Life Technologies | Catalog number: A20000 | |
| Chemical compound, drug | n-Decyl-β-D-Maltopyranoside | anatrace | D322LA | |
| Software, algorithm | cryoSPARC | Structura Biotechnology | RRID:SCR_016501 | |
| Software, algorithm | RELION | *Scheres, 2012* | https://www3.mrc-lmb.cam.ac.uk/relion/index.php/Main_Page | |
| Software, algorithm | Chimera | *Pettersen et al., 2004* | RRID:SCR_004097 | |
| Software, algorithm | PyMOL | Schrodinger | RRID:SCR_000305 | |
| Software, algorithm | PHENIX | *Adams et al., 2010* | RRID:SCR_014224 | |
| Software, algorithm | COOT | *Emsley et al., 2010*; | RRID:SCR_014222 | |
| Software, algorithm | PRISM | GraphPad | RRID:SCR_002798 | |

*Continued on next page*

*Continued*

| Reagent type (species) or resource | Designation | Source or reference | Identifiers | Additional information |
|---|---|---|---|---|
| Software, algorithm | CAVER | | https://caver.cz/ | |
| Software, algorithm | NAMDINMATOR | *Kidmose et al., 2019* | https://namdinator.au.dk/ | |
| Software, algorithm | MAXCHELTOR | *Bers et al., 1994* | https://somapp.ucdmc.ucdavis.edu/pharmacology/bers/maxchelator/webmaxc/webmaxcS.htm | |
| Software, algorithm | MO.Affinity Analysis software v2.3 | Nanotemper | | |
| Software, algorithm | CHARMM c42a2 with PHMD | *Brooks et al., 2009*; *Khandogin and Brooks, 2005* | https://charmm-gui.org | |
| Software, algorithm | CpHMD-Analysis | *Huang et al., 2021 Wallace and Shen, 2011* | https://github.com/Hendejac/CpHMD-Analysis | |
| Software, algorithm | CHARMM-GUI | *Jo et al., 2008*; *Jo et al., 2009*; *Lee et al., 2016* | | |
| Software, algorithm | GROMACS 2021.1 | *Abraham et al., 2015* | | |
| Software, algorithm | Multibind | *Kenney and Beckstein, 2023* | https://github.com/Becksteinlab/multibind; *Beckstein and Kenney, 2023* | |
| Others | Monolith standard treated Capillaries | Nanotemper | K022 | https://shop.nanotempertech.com/en/monolith-capillaries-1000-count-17 |
| Others | C-Flat 1.2/1.3-4Cu-50 grids | Protochips, Inc | | https://www.emsdiasum.com/c-flat-family-of-holey-carbon-gold-grids-for-stem-and-cryo-tem |

## Protein expression and purification

YiiP was expressed in *E. coli* (BL21(DE3)-CodonPlus-RIPL) from a pET vector that included an N-terminal decahistidine tag. Cells were grown in LB media supplemented with 30 µg/ml kanamycin at 37 °C until they reached an $OD_{600}$ of 0.8. After cooling the media to 20 °C, expression was induced by addition of 0.5 mM isopropyl-β-D-thiogalactoside followed by overnight incubation at 20 °C. Cells were harvested by centrifugation at 4000x*g* for 1 h, resuspended in lysis buffer (20 mM HEPES, pH 7.5, 100 mM NaCl, 10% glycerol, and 500 µM tris(2-carboxyethyl)phosphine) - 100 ml of buffer per 20 g of cells - and then lysed with a high-pressure homogenizer (Emulsiflex-C3; Avestin, Inc Ottawa Canada). Protein was extracted from the membrane by adding 1.5 g dodecyl-β-D-maltoside per 100 ml of cell lysate followed by 2 hr of incubation at 4 °C. Insoluble material was removed by centrifugation at 100,000x*g* for 30 min. The supernatant was loaded onto a Ni-NTA affinity column pre-equilibrated in buffer A (20 mM HEPES, pH 7.5, 100 mM NaCl, 10% glycerol, and 0.05% dodecyl-β-D-maltoside). The column was washed by addition of buffer A supplemented with 20 mM imidazole and protein was then eluted using a gradient of imidazole ranging from 20 to 500 mM. Peak fractions were combined, supplemented with tobacco etch virus (TEV) protease (1:10 weight ratio of TEV:YiiP) to cleave the decahistidine tag, and dialyzed overnight at 4 °C against buffer A. TEV protease was removed by loading the dialysate onto an Ni-NTA column and collecting the flow-through fractions. After concentration, a final purification was done with a Superdex 200 size-exclusion chromatography (SEC) column (GE Healthcare, Chicago, Illinois) equilibrated with SEC buffer (20 mM HEPES, pH 7.5,150 mM NaCl, 0.2% n-decyl-β-D-maltoside, and 1 mM tris(2-carboxyethyl)phosphine).

Fab selection, modification, expression, and purification has been described previously (*Lopez-Redondo et al., 2021*). Briefly, the construct designated Fab2r was expressed in *E. coli* strain 55244 from a freshly prepared transformation. Cells were cultured for ~24 hr at 30 °C with constitutive expression behind an innately leaky T4 promoter. Cell pellets were harvested by centrifugation, lysed with a high-pressure homogenizer and Fab was purified with a 5 ml HiTrap Protein G HP column (GE Healthcare). Pooled fractions were dialyzed against sodium carbonate buffer (pH 5.0) overnight at 4 °C and further purified with Resource-S cation exchange column (GE Healthcare). Finally, pure Fab protein was pooled and dialyzed against SEC buffer.

## Cryo-EM sample preparation and structural analyses

To ensure that YiiP was fully loaded with $Zn^{2+}$, the purified protein was initially incubated with metal ion chelators (0.5 mM EDTA and 0.5 mM N,N,N′,N′-tetrakis(2-pyridinylmethyl)–1,2-ethanediamine [TPEN]) for 24 hr at 4 °C to remove ions that co-purified with the protein, as seen in previous work (*Lopez-Redondo et al., 2021*). The sample was then dialyzed against YiiP SEC buffer containing 0.5 mM EDTA for 10 hr to eliminate chelator-metal complexes. Finally, YiiP was loaded with $Zn^{2+}$ by dialysis (10 hr with four buffer exchanges) against SEC buffer supplemented with 0.25 mM $ZnSO_4$. YiiP was then incubated with Fab2r at a 1:1 molar ratio for 1 hr at 20 °C to form the YiiP-Fab2r complex. This complex was purified by SEC using a Superdex 200 column equilibrated with SEC buffer supplemented with 0.25 mM $ZnSO_4$. Peak fractions at 3–5 mg/ml were used immediately for preparation of cryo-EM samples. Specifically, 3–4 μl were added to glow-discharged grids (C-Flat 1.2/1.3-4Cu-50; Protochips, Inc) that were blotted under 100% humidity at 4 °C and plunged into liquid ethane using a Vitrobot (Thermo Fisher Scientific, Inc Bridgewater NJ).

The YiiP-Fab2r complex was imaged with a Titan Krios G3i electron microscope (Thermo Fisher Scientific, Inc) equipped a Bioquantum energy filter and K2 or K3 direct electron detector (Gatan, Inc Pleasanton CA) with a pixel size of ~1 Å and a total dose of ~50 electrons/$Å^2$. Micrographs containing crystalline ice, excessive contamination, or imaging artifacts were removed and the resulting micrographs were imported into cryoSPARC v2.15 (*Punjani et al., 2017*) for analysis. Particles were picked based on templates generated in previous work and an initial set of particles were subjected to two rounds of 2-D classification to remove false positives. The resulting particles were then subjected to successive rounds of ab initio reconstruction with C1 symmetry and a resolution cutoff starting at 12 Å and declining to 8 Å; at each step, the best of two output classes was carried forward to the next round. The resulting particles were used for heterogeneous refinement against two or three reference structures derived from the ab initio jobs, still with C1 symmetry. A final selection of particles was then used for non-uniform refinement using both C1 and C2 symmetry. Postprocessing steps included calculation of local resolution and evaluation of 3-D variability (*Punjani and Fleet, 2021*). For the D70A construct, RELION (*Scheres, 2012*) was used to process a class of particles corresponding to a dimer of dimers (*Figure 1—figure supplement 3*). After exporting these particles from cryoSPARC, steps of 2D classification and ab initio reconstruction were repeated in RELION, followed by 3D refinement of ~117,000 dimer-of-dimer particles with C2 symmetry. A mask encapsulating one dimer was created with Chimera (*Pettersen et al., 2004*) and symmetry expansion followed by signal subtraction was used to generate a new set with ~234,000 dimeric particles. This expanded particle set was imported back into cryoSPARC and combined with the class of isolated dimer particles. This combined particle set was used for hetero-refinement to segregate dimer complexes with symmetrical and asymmetrical TMD's. These segregated particle sets were then used for final non-uniform refinement.

For model building of symmetrical structures from WT, D287A, D51A, we started with the deposited coordinates from wild type (PDB, 7KZZ). This model was docked as a rigid body to the map and adjusted manually in Coot (*Emsley et al., 2010*) as a starting structure for refinement using PHENIX (*Adams et al., 2010*). For D70A and D287A/H263A mutants, the CTD and the TMD from either 7KZZ or 7KZX were separately docked to the map. After adjustment and crude rebuilding in Coot (*Emsley et al., 2010*), these models were submitted to NAMDINATOR for Molecular Dynamics Flexible Fitting (*Kidmose et al., 2019*). The resulting models were used as starting points for PHENIX refinement. This refinement consisted of multiple rounds that alternated between real-space refinement and manual adjustment using Coot. Atomic models were then displayed using CHIMERA (*Pettersen et al., 2004*) and the PyMOL Molecular Graphics System (Schrödinger, LLC, New York NY).

For characterization of cavities running from site A to the cytoplasm, we used Caver Analyst 2.0 Beta (*Jurcik et al., 2018*). The starting point was set near D51 in models for WT and D70A_asym and we selected the shortest tunnel that ran to the cytoplasm. Settings used for the calculations were as follows. Approximation: 12, Minimum probe radius: 0.9, Clustering threshold: 3.5, Shell depth: 2.0, Shell radius: 3.0. $Zn^{2+}$ ions were excluded from the calculation.

## MST experiments

YiiP mutants were labelled with Alexa fluor 488 fluorescent dye (Invitrogen life Technologies, Carlsbad CA) by adding 2.5 µL of dye from a 16 mM stock solution in DMSO to 200 µL of protein at 1–2 mg/ml in SEC buffer supplemented with 10% glycerol at pH 7. The reactive group of this dye was N-hydroxy-succinimide, which at pH 7 preferentially targets the N-terminus of the polypeptide. Although labeling of lysine residues is also possible, we believe this was minimal due to the low labeling stoichiometries of ~1:1 used for our experiments. $Zn^{2+}$ was removed by adding 0.5 mM EDTA and 0.5 mM TPEN followed by overnight incubation at 4 °C. Excess dye and chelated $Zn^{2+}$ were removed using several cycles of dilution with SEC buffer and concentration with a 50 kDa cutoff concentrator (AMICON, Millipore Sigma, Burlington MA). For titration, $Zn^{2+}$ was buffered either by 0.5 mM sodium citrate or 0.2 mM NTA; the total amount of added $ZnSO_4$ was varied to achieve the desired concentration of free $Zn^{2+}$, according to the program MAXCHELATOR (*Bers et al., 1994*). Protein concentration during the titration varied from 8 to 100 nM. After 1:1 mixing of protein and $Zn^{2+}$ solutions, samples were centrifuged for 5 min at 2000xg, then loaded into standard treated capillaries for measurement with a Monolith NT.115 MST instrument (NanoTemper Technologies, South San Francisco, CA); measurements were taken at 37 °C with LED power ranges from 20–60% and medium MST power. Data from three independent titrations were analyzed with the MO.Affinity Analysis software v2.3 using MST on-time of 15 s. For determination of $K_d$, data was fitted with a curve based on the law of mass action,

$$F\left(C_{Zn}\right) = F_u + \frac{\left(F_b - F_u\right) * \left(C_{Zn} + C_P + K_d - \sqrt{\left(C_{Zn} + C_P + K_d\right)^2 - 4 * C_P * C_{Zn}}\right)}{2 * C_P}$$

where $F(C_{Zn})$ is the fraction bound, $C_{Zn}$ is concentration of free zinc, $C_P$ is the concentration of YiiP protein, $F_b$ and $F_u$ refer to the normalized fluorescence in the bound and unbound state, and $K_d$ is the affinity constant. Alternatively, the data was fit with the Hill equation to assess cooperativity,

$$F\left(C_{Zn}\right) = F_u + \frac{\left(F_b - F_u\right)}{1 + \left(\frac{EC_{50}}{C_{Zn}}\right)^n}$$

where $EC_{50}$ is the half-maximal effective concentration (akin to $K_d$) and $n$ is the Hill coefficient.

## Overview of MD simulations

Two types of all-atom, explicit solvent MD simulations were conducted to assess the effects of proton and zinc binding on the structure of the YiiP dimer in a lipid bilayer. We used fixed charge equilibrium MD simulations (using a force field with unchanging parameters) to investigate the interactions

**Table 5.** MD simulations.

| Simulation | Length | N of replicas | Total simulation time | Software |
|---|---|---|---|---|
| Fixed charge *apo*[*] | 1000 ns | 3 | 3000 ns | GROMACS |
| Fixed charge *holo*[*] | 1000 ns | 3 | 3000 ns | GROMACS |
| Fixed charge *empty site B* | 1000 ns | 6 | 6000 ns | GROMACS |
| Fixed charge D72A | 1000 ns | 3 | 3000 ns | GROMACS |
| CpHMD apo | 12 ns | 30 (pHREX) | 360 ns | CHARMM |

[*]Fixed charge" simulations were repeated three times. The CpHMD simulations were performed as coupled pH-replica exchange simulations. *Simulations for apo and holo YiiP were taken from our previous work (*Lopez-Redondo et al., 2021*).

between $Zn^{2+}$ ions and YiiP. We also carried out replica exchange constant pH MD (CpHMD) simulations to calculate the p$K_a$ values of all titratable residues and assess the microscopic protonation states of the ion binding sites in the absence of $Zn^{2+}$. Unlike CpHMD simulations, fixed charge simulations do not allow for dynamic protonation or deprotonation of residues. Despite this limitation, they are our preferred approach to study protein-zinc ion interactions and conformational changes because they can be run more efficiently than CpHMD and were thus used to quantify the effect of zinc ions on the structure of YiiP (see *Table 5* for a summary of these simulations). The zinc ions in the system were simulated with the non-bonded dummy model described in our previous work *Lopez-Redondo et al., 2021*; parameter files are available as package 2934 in the Ligandbook repository (*Domanski et al., 2017*; https://ligandbook.org/package/2934). Briefly, parameters governing this model were refined based on experimental hydration free energy, ion-oxygen distance and coordination number of the water in the first hydration shell, ultimately reproducing these values with errors of 1% and 0.3%, respectively. The model was further validated using simulations of known zinc-binding proteins, in which stability of the protein and the geometry of the binding complexes were well maintained, though coordination distances were slightly longer than experimental values, especially for sulfur atoms from cysteine residues (*Lopez-Redondo et al., 2021*). Given that YiiP employs aspartate or histidine residues for $Zn^{2+}$ coordination, the model is well-suited for the current work.

## Fixed charge equilibrium MD simulations

MD simulations in the *holo* (with $Zn^{2+}$ in A, B, and C sites) and *apo* (no $Zn^{2+}$) state were taken from our previous study (*Lopez-Redondo et al., 2021*). To investigate the influence of the binding site B, we generated an '*empty site B*' structure by removing the $Zn^{2+}$ ions from site B in both protomers of PDB ID 5VRF, the same starting structure used previously. In order to study the influence of the salt-bridge D72-R210, simulations were also conducted after applying the D72A mutation to the holo structure (PDB ID 5VRF). In these D72A simulations, $Zn^{2+}$ ions were bound in all sites (A, B, C1, C2), that is D72A simulations differed from the holo simulations only in the mutation.

The membrane-protein systems were modeled by embedding the YiiP dimer into a 4:1 palmitoyloleoylphosphatidylethanolamine:palmitoyloleoylphosphatidylglycerol (POPE:POPG) lipid bilayer, which approximates the composition of the plasma membrane from *E. coli* (*Raetz, 1986*), and solvated the system with water as well as sodium and chloride ions corresponding to a concentration of 100 mM using CHARMM-GUI v1.7 (*Jo et al., 2008*; *Jo et al., 2009*; *Lee et al., 2016*). We used GROMACS 2021.1 (*Abraham et al., 2015*) with the CHARMM36 force field, the CMAP correction for proteins (*MacKerell et al., 1998*; *Mackerell et al., 2004*), CHARMM36 lipids (*Klauda et al., 2010*), and the CHARMM TIP3P water model. The *empty site B* system contained 117,512 atoms, the apo system contained 117,394 atoms, and the D72A system contained 117,418 atoms. All systems were constructed in a hexagonal simulation cell with initial dimensions 101 Å × 101 Å×135 Å. Default protonation states of all ionizable residues were used based on the experimental pH of 7. The neutral HSD tautomer (proton on the $N_\delta$) was selected to model all histidines except H73 and H155, which were modeled with HSE (proton on the $N_\varepsilon$) based on their orientation relative to $Zn^{2+}$ ions in the cryo-EM structure PDB ID 5VRF.

The systems were first energy minimized and underwent a 3.75-ns six-stage equilibration procedure with position restraints on protein and lipids, following the CHARMM-GUI protocol (*Jo et al., 2008*). Three copies of 1-μs production simulations were carried out with *empty site B*, starting from the same initial system conformation but with different initial velocities. All simulations were performed under periodic boundary conditions at constant temperature ($T$=303.15 K) and pressure ($P$=1 bar). The velocity rescaling thermostat (*Bussi et al., 2007*) was used to maintain the temperature with a time constant of 1 ps and separate temperature-coupling groups for protein, lipids, and solvent. A semi-isotropic pressure coupling scheme was implemented using the Parrinello-Rahman barostat (*Parrinello and Rahman, 1981*) with a time constant of 5 ps, a compressibility of $4.6 \times 10^{-5}$ bar$^{-1}$. Long-range electrostatics were calculated with the smooth particle mesh Ewald method (*Essmann et al., 1995*) under tinfoil boundary conditions with an initial cutoff of 1.2 nm, which was optimized during the simulation, and interactions beyond the cutoff were calculated in reciprocal space with a fast-Fourier transform on a grid with spacing 0.12 nm and fourth-order spline interpolation. The van der Waals interactions were switched smoothly to 0 between 1.0 nm and 1.2 nm, and the interactions were shifted over the whole range and reduced to 0 at the cutoff.

The Verlet neighbor list was updated dynamically by GROMACS for optimized performance with a buffer tolerance of 0.005 kJ/mol/ps. Bonds to hydrogen atoms were treated as rigid holonomic constraints with the P-LINCS algorithm (*Hess, 2008*) with an expansion order of four and two LINCS iterations; alternatively, SETTLE (*Miyamoto and Kollman, 1992*) was used for water molecules. The classical equations of motion were integrated with the leapfrog algorithm with a time step of 2 fs.

## CpHMD simulations with pH replica exchange

In conventional MD simulations, the protonation states of titratable groups in the system are fixed. To investigate the role of protons in the $Zn^{2+}$ transport of YiiP, we performed membrane-enabled hybrid-solvent continuous constant pH MD simulations (*Huang et al., 2021*). The currently available implementation of the membrane-enabled CpHMD method does not take into account the direct effect of ions on the titration of nearby residues so we only ran CpHMD simulations for the apo system.

CpHMD simulations were initialized with the apo model (PDB ID 5VRF with $Zn^{2+}$ ions removed). An initial 250-ns production simulation was run for the CpHMD apo system to fully relax the membrane, using the same GROMACS 2021.1 protocol described above for fixed charge equilibrium simulations.

The membrane-enabled hybrid-solvent continuous CpHMD simulations were performed using the CHARMM program version c42a2 (*Brooks et al., 2009*) with the PHMD module (*Khandogin and Brooks, 2005*; *Lee et al., 2004*) and the pH replica-exchange (REPDSTR) module (*Wallace and Shen, 2011*). The conformations of YiiP were sampled using conventional all-atom simulations with the CHARMM22/CMAP all-atom force field (*MacKerell et al., 1998*; *Mackerell et al., 2004*), the CHARMM36 lipid force field (*Klauda et al., 2010*), and the CHARMM modified TIP3P water model. The titration coordinates were propagated using the membrane GBSW implicit-solvent model (*Im et al., 2003*) with the GBSW input radii for the protein taken from *Chen et al., 2006*. Based on the average distance between the C2 atoms of the lipids in the cytoplasmic- and periplasmic-facing leaflets, the thickness of the implicit bilayer was set to 40 Å with a switching distance of 5 Å for the transition between the low dielectric slab and bulk solvent. The implicit membrane was excluded by two cylinders with a radius of 14 Å placed at the center of mass of each protomer of YiiP. The radius was selected to maximize the coverage of the interior of the protein with minimal overlapping of the implicit membrane.

The final snapshot of the equilibrium simulation was used as the initial structure for the CpHMD simulation. Dummy hydrogen atoms were added to the carboxylate groups of acidic residues following the documentation of the PHMD module (*Wallace and Shen, 2011*) in CHARMM (*Brooks et al., 2009*) using the HBUILD facility. The system was then equilibrated with energy minimization using 50 steps of steepest descent followed by 50 steps of adopted basis Newton-Raphson algorithms and CpHMD at pH 7 for 1 ns, whereby the harmonic restraints on the protein heavy atoms were reduced from 1 kcal·mol$^{-1}$Å$^{-1}$ to zero.

The production simulation was then performed using hybrid-solvent CpHMD with the pH replica-exchange protocol (*Huang et al., 2021*; *Wallace and Shen, 2011*), using 30 replicas with pH ranging from 1.5 to 11.5. The specific pH conditions were 1.5, 1.75, 2, 2.5, 2.75, 3, 3.25, 3.5, 3.75, 4, 4.25, 4.5, 4.75, 5, 5.25, 5.5, 6, 6.5, 7, 7.5, 8, 8.5, 8.75, 9, 9.5, 10, 10.25, 10.5, 11, 11.5, chosen to ensure that the exchange rate between the nearby replicas was higher than 0.2. Each replica was simulated under periodic boundary conditions at constant temperature ($T$=303.15 K), pressure ($P$=1 bar), and specified pH. A modified Hoover thermostat method (*Hoover, 1985*) was used to maintain the temperature, while pressure was maintained using the Langevin piston coupling method (*Feller et al., 1995*) with a piston mass of 2500 amu. Long-range electrostatics were evaluated with the particle mesh Ewald method (*Darden et al., 1993*) with a real-space cutoff of 1.2 nm, and interactions beyond the cutoff were calculated in reciprocal space with a fast-Fourier transform on a grid with 0.09 nm spacing and sixth-order spline interpolation. The Lennard–Jones forces were switched smoothly to 0 between 1.0 and 1.2 nm, and the potential was shifted over the whole range and reduced to 0 at the cutoff. Bonds to hydrogen atoms were constrained with the SHAKE algorithm to allow a 2 fs time step. To avoid a spike in potential energy due to a lack of solvent relaxation (*Wallace and Shen, 2011*), a GBSW calculation was executed every 5 MD steps to update the titration coordinates. An attempt to exchange adjacent pH replicas was made every 1000 MD steps (corresponding to 2 ps). Each replica simulation lasted 12 ns for a total aggregate sampling time of 360 ns. Many replicas exchanged across a large

fraction of available pH space (*Figure 4—figure supplement 2*), indicating that 12 ns per replica were sufficient for sampling the degrees of freedom near the protonation sites.

## Analysis of fixed charge MD simulations

Analysis of the trajectories was carried out with Python scripts based on MDAnalysis (*Gowers et al., 2016*). RMSDs of $C_\alpha$ atoms of the whole protein, TMD, and CTD were calculated using the qcprot algorithm (*Liu et al., 2010*) after optimally superimposing the structure on the same $C_\alpha$ atoms of the cryo-EM structure. Similarly, root mean square fluctuation (RMSF) of $C_\alpha$ atoms of the whole protein, TMD, and CTD were calculated. To assess the relative motion between the two domains, a CTD-TMD rotation angle was calculated from the rotation matrix which minimized the RMSD of the CTD after superimposing the protein on the TMD domain of the reference structure (PDB ID 5VRF), as in our previous work (*Lopez-Redondo et al., 2021*). The existence of the salt-bridge Asp72-Arg210 was quantified using the shortest distance between $O_\delta$ atoms of Asp72 and hydrogen atoms on the side chain of Arg210.

## Analysis of CpHMD simulations

The titration coordinates were extracted from CpHMD output files as time series $S(t)$ (*Figure 4—figure supplement 3*) with the CpHMD-Analysis scripts (https://github.com/Hendejac/CpHMD-Analysis) (*Henderson, 2021*). The primary purpose of these simulations is to obtain microscopic $pK_a$ values for titratable residues. We first describe the conventional approach to obtain per-residue $pK_a$ values using the heuristic generalized Hill equation and then in the next section demonstrate an alternative inference approach based directly on statistical mechanics.

The deprotonation fraction $S$ of a titratable site was calculated from the titration time series (*Figure 4—figure supplement 3*) as the number of trajectory frames $N$ of the residue in the deprotonated and protonated states as

$$S = \frac{N_{\text{deprot}}}{N_{\text{deprot}} + N_{\text{prot}}}$$

where the site is identified as deprotonated when the CpHMD titration coordinate $\lambda$ is greater than 0.8 and protonated when the titration coordinate is less than 0.2. Individual residue $pK_a$'s were obtained from a fit to the generalized Henderson-Hasselbalch equation (Hill equation) (*Henderson et al., 2020*; *Huang et al., 2016*; *Huang et al., 2021*) for the deprotonated fraction $S$ as a function of pH,

$$S(\text{pH}) = \frac{1}{1 + 10^{n(pK_a - \text{pH})}}$$

where $n$ is the Hill coefficient, which represents the slope of the response curve (*Figure 4—figure supplement 4*). The mean $pK_a$ of a residue was calculated for each pair of equivalent residues in protomer A and protomer B because these residues were sampled independently in the CpHMD simulation. The statistical error was estimated as the absolute difference between the $pK_a$'s of protomer A and B residues and their mean value.

The CpHMD simulations track the protonation of every single titratable residue and thus provide detailed microscopic information on the exact protonation state for each binding site. The distribution of these microscopic states named S0 to S15 for site A (*Figure 4—figure supplement 5*, inset table) and S0 to S7 for site B (*Figure 4—figure supplement 6*, inset table) forms the basis for our alternative calculation of microscopic $pK_a$ values. In order to obtain these CpHMD protonation state distributions, we treated titration coordinate data $S(t)$ from protomer A and B as independent; thus, by concatenating them we effectively doubled our sampling. Using the same criterion for bound/unbound protons and the definition of microstates for site A (*Figure 4—figure supplement 5*) and site B (*Figure 4—figure supplement 6*) we generated separate microstate time series for sites A and B. These time series were then histogrammed to derive the distributions of the microscopic protonation states of site A (*Figure 4—figure supplement 5A*) and B (*Figure 4—figure supplement 6A*).

## Calculation of microscopic, state-dependent p$K_a$ values with the *Multibind* method

The *Multibind* method generates thermodynamically consistent models for systems of coupled reactions by using a maximum likelihood approach to combine kinetic or thermodynamic measurements from different sources (*Kenney and Beckstein, 2023*). Given free energy differences between states (either from simulations or experiments), *Multibind* generates a complete set of free energies for all states in the form of a potential graph that obeys path-independence of free energies and detailed balance while being maximally consistent with the input data. From this set of free energies, all macroscopic thermodynamic observables can be calculated. Without such an approach, thermodynamically inconsistent models arise due to random errors in the input measurements. For $Zn^{2+}$ and proton binding, the free energy differences in the potential graph are calculated as functions of the external parameters, namely the free $Zn^{2+}$ concentration [X] and the pH, and with p$K_a$ and $Zn^{2+}$ standard state binding free energy values as input. The binding free energy of a reaction $A + X \rightleftharpoons A : X$ is calculated as

$$\beta \Delta G_{\text{bind}} \left( [X] \right) = \ln \left( \frac{K_D}{[X]} \right) = \ln \left( \frac{K_D}{c_0} \right) - \ln \left( \frac{[X]}{c_0} \right) = \beta \Delta G_{\text{bind}}^0 - \ln \left( \frac{[X]}{c_0} \right)$$

where $\beta = \frac{1}{k_B T}$ ($k_B$ is Boltzmann's constant), $K_D$ is the dissociation constant, and the binding free energy is written as a sum of $\Delta G^0_{\text{bind}}$, the binding free energy at the standard state concentration $c_0 = 1$ M, and a term depending on the ligand concentration [X]. For proton binding, the free energy difference is expressed equivalently as

$$\beta \Delta G_{\text{prot}} \left( \text{pH} \right) = \ln \left( 10 \right) \left( \text{pH} - \text{p}K_a \right),$$

where p$K_a$ is the acid dissociation constant.

We used the Python implementation of *Multibind* (https://github.com/Becksteinlab/multibind) (*Kenney and Beckstein, 2023* ) to construct thermodynamic models for binding sites A and B, with the assumption that all binding sites across both protomers are independent. Each model describes the transitions between all possible states as either binding of a proton or binding of a $Zn^{2+}$ ion. With four titratable residues, site A has $M=2^4 = 16$ $Zn^{2+}$-free protonation microstates (states S0 to S15 in *Figure 4—figure supplement 5*, inset table) and 16 $Zn^{2+}$-bound protonation microstates for a model with 32 states in total (*Figure 4h*). Site B has three titratable residues and thus $M=8$ for a model with 16 states in total (states S0 to S7 in *Figure 4—figure supplement 6*, inset table). Because only free energy differences between states are measurable, we can arbitrarily specify one state as a reference with zero free energy. Here, we chose the 0-proton unbound state S0 as the reference state with $G_0=0$, but calculated observables are independent of this particular choice. The probability of observing the system in the microstate $i$ is

$$P_i \left( \text{pH}, \left[ Zn^{2+} \right] \right) = \frac{1}{Z} e^{-\beta G_i}$$

where $G_i$ is the free energy of any state $i$ and $Z$ is the partition function

$$Z \left( \text{pH}, \left[ Zn^{2+} \right] \right) = \sum_{k=1}^{M} e^{-\beta G_{k,\text{unbound}}} + \sum_{k=1}^{M} e^{-\beta G_{k,\text{Zn}}}$$

which contains the partition functions of the unbound and bound states. We note that the microstate probabilities $P_i$ can also be directly obtained from the CpHMD simulations, which enables us to use *Multibind* as an inverse method to infer microscopic p$K_a$ from CpHMD data (without resorting to the generalized Hill equation fit), as described below.

The fraction of YiiP protomers that have a $Zn^{2+}$ ion bound to the binding site in question, the *bound fraction*, is

$$\langle X \rangle = Z^{-1} \sum_{k=1}^{M} e^{-\beta G_{k,\text{Zn}}}$$

**Table 6.** MST convergence criteria.

| Target | Binding site | RMSD cutoff |
| --- | --- | --- |
| CpHMD microstate probability distributions | Site A | 0.0382 |
| CpHMD microstate probability distributions | Site B | 0.0447 |
| MST bound fraction | Site A | 0.0760 |
| MST bound fraction | Site B | 0.0995 |

The bound fraction can be obtained experimentally from MST measurements.

With the ability to calculate microstate probability or bound fraction at specific pH or $Zn^{2+}$ concentration, we devised an inverse approach to *infer* microscopic, state-dependent $pK_a$'s and standard binding free energies ($\Delta G^0_{bind}$) from either CpHMD data (to avoid using the Hill equation with the deprotonated fractions, which are aggregated over multiple states and may therefore mask coupling between residues) or from experimental MST measurements.

The inverse *Multibind* approach generates successive microscopic models with a Monte Carlo (MC) scheme and compares the calculated observables to the target values. Observables correspond to microstate probability distributions $P_i$(pH) for CpHMD or bound fraction <X > as a function of pH and $Zn^{2+}$ concentration for MST. Once convergence is reached, the microscopic model contains the inferred microscopic state-dependent $pK_a$ values and $Zn^{2+}$ binding free energies. We implemented a simple finite-temperature MC algorithm that minimizes the RMSD between the given and computed macroscopic features. The model was initialized with $pK_a$ values from CpHMD (obtained with the Hill equation) or $\Delta G^0_{bind}$ binding free energies calculated from the experimental $K_d$ from the MST analysis. For each MC step, the target observable was calculated for a range of pH values and $Zn^{2+}$ concentrations. For as long as the RMSD between the computed and the target observable remained above a cutoff (see *Table 6*), a new set of $pK_a$ or $\Delta G^0_{bind}$ was generated by adding a random value drawn from a uniform distribution ranging from –0.2 to 0.2 $pK_a$ units (for $pK_a$) or –0.2 to 0.2 $k_BT$ (for binding free energies) to all microscopic parameters. A new set of changes was accepted with probability (Metropolis criterion)

$$P(pK_a, \Delta G^0_{bind} \to pK_a', \Delta G^0_{bind}') = min(1, e^{-\frac{\Delta RMSD}{r}})$$

where $\Delta RMSD$ is the difference between the RMSDs between the target and the computed observable for the old set of $pK_a$'s or $\Delta G^0_{bind}$'s and the new set

$$\Delta RMSD = RMSD\left(pK_a', \Delta G^0_{bind}'\right) - RMSD\left(pK_a, \Delta G^0_{bind}\right)$$

The fictional temperature in the Metropolis criterion, $r$, was set to 0.0001 to be on the order of typical per-step changes. The RMSD cutoffs (*Table 6*) were chosen from initial runs using a zero-temperature MC algorithm, which accepted a new set of changes only when $\Delta RMSD <0$. These initial runs were stopped after 2000 steps and after no changes had been accepted in the last 100 or more steps. The zero-temperature MC ensured that the final set generated the smallest RMSD, while the finite-temperature MC used in the production runs allowed larger parameter space to be sampled. Fifty runs were performed for each target. Averaged $pK_a$ values and corresponding standard deviations were calculated from the 50 independent MC runs. Although these standard variations of our parameter estimates weakly depend on the somewhat arbitrary choice of $r$, we chose to report them instead of a rigorous statistical error (which is unavailable with the current approach) to provide a sense of variability of the estimates.

For site B, the CpHMD simulations suggested that H73 and H77 behaved identically and had almost the same microscopic $pK_a$ values. We calculated the 'coupling energy' (*Ullmann, 2003*) between H73 and H77,

$$W = pK_a\left(S0 \to S2\right) - pK_a\left(S1 \to S3\right) = pK_a\left(S0 \to S1\right) - pK_a\left(S2 \to S3\right)$$

(expressed in $pK_a$ units, i.e. energy divided by $k_B T \ln 10$), from the microscopic CpHMD state $pK_a$'s (see source data for *Figure 4—figure supplements 5 and 6*). The coupling energy measures how the protonation of a specific residue (e.g. H73) depends on the protonation of another residue (here, H77). The coupling energy is the difference in $pK_a$ when H77 remains deprotonated while H73 binds a proton (S0→S2, see *Figure 4—figure supplement 6*) compared to the situation when H77 is already protonated (S1→S3). The H73-H77 coupling energy for the CpHMD simulation was $W=+1.0$ for D70 deprotonated and $W=+0.2$ for D70 protonated. Because $W>0$, binding of a proton to one of the histidines decreases the $pK_a$ of the other, thus decreasing proton binding via anti-cooperative coupling. Preliminary calculations with the MST data, however, showed that small initial differences in the starting values could lead to large $pK_a$ shifts in H73 and H77, with the directions determined by the initial ordering of $pK_a$ values. Because we had no specific evidence that the two histidines should behave differently, we considered the behavior of the initial unconstrained MST inference calculations to be problematic. This problem indicated that the inverse approach can be sensitive to initial values and that there may not always be sufficient target data to constrain the microscopic model. We therefore treated the two residues as symmetrical, that is both should be behaving in the same way, and imposed a constraint in the *Multibind* approach so that both were assigned the same $pK_a$ at each step, thus effectively imposing the coupling observed in the CpHMD simulations. The interaction energies between H73 and H77 for the MST-inference data are $W=+6.2$ (D70 deprotonated) and $W=-0.70$ (D70 protonated). The dominant state is the one with D70 deprotonated due to the low $pK_a$ of D70 and hence the overall behavior of H73 and H77 remains strongly anti-cooperative in the MST-inference model.

For Site A, the initial CpHMD simulations did not indicate any degeneracy in $pK_a$ values such as the one for H73/H77 and so no additional constraints were applied for any of the Site A MST-inference calculations. We also calculated the coupling energy for H155 and D159 in site A from the microscopic $pK_a$ values (see source data for *Figure 4—figure supplements 5 and 6*) for the CpHMD and the MST-inference models for (1) no protons bound to D51 and D47 ($W_{MST}=-5.05$, $W_{CpHMD} = +1.26$), (2) D51 protonated ($W_{MST}=-7.39$, $W_{CpHMD} = +0.59$), (3) D51 protonated ($W_{MST}=-5.26$, $W_{CpHMD} = +1.02$), and (4) both D47 and D51 protonated ($W_{MST}=-2.39$, $W_{CpHMD} = +1.79$). Thus, refinement against the experimental MST data changes the model based on the CpHMD results from anti-cooperative binding ($W>0$) to strong cooperative binding ($W<0$).

The *inverse* Multibind approach has some limitations when applied to binding curves such as the MST data. The binding isotherms, even when covering a range of pH values, do not contain enough data to determine the microscopic model fully. Therefore, it was necessary to use the CpHMD $pK_a$ values to initialize the model instead of arbitrary starting values. Thus, in the current implementation, the MST inference approach should be viewed as a refinement process for the simulation-derived parameters, restricted or guided by the experimental data. The resulting thermodynamic model allows the calculation of state probabilities (and any other properties) at pH values outside the experimental range but it should be noted that these calculated quantities are extrapolations that may not be accurate due to the lack of experimental data to constrain the model; for instance, conformational changes may occur that could drastically alter the interactions but these changes would not have been captured in a pure binding model such as the one in *Figure 4H*.

An effective $pK_a$ under the $Zn^{2+}$-free condition was estimated for each residue by fitting the deprotonation fractions (generated from the microscopic *Multibind* model) to the Hill-Langmuir equation. The deprotonation fraction of a specific residue was calculated by summing up the probabilities of microscopic states where the residue was deprotonated. The $Zn^{2+}$ concentration was set to $10^{-20}$ M to approximate the $Zn^{2+}$-free condition. For the symmetrized residues H73 and H77, the Hill-Langmuir equation did not produce a satisfactory fit. In order to compute macroscopic $pK_a$'s for such a coupled system, we followed previous work (*Henderson et al., 2020*; *Ullmann, 2003*) and fitted the total number of protons bound to the two residues to the so-called 'coupled titration model'

$$N_{prot} = 2 - \left[ S_1 \left( \text{pH} \right) + S_2 \left( \text{pH} \right) \right] = \frac{10^{\left( pK_2 - \text{pH} \right)} + 2 \times 10^{\left( pK_1 + pK_2 - 2\text{pH} \right)}}{1 + 10^{\left( pK_1 - \text{pH} \right)} + 10^{\left( pK_1 + pK_2 - 2\text{pH} \right)}}$$

where $N_{prot}$ is the total number of protons, $S_1$ and $S_2$ are the deprotonation fractions of the two residues, and $pK_1$ and $pK_2$ are the two coupled $pK_a$'s describing the binding of the first proton and the second proton to the coupled titrating sites. These two $pK_a$'s can be interpreted as effective $pK_a$ values of two uncoupled residues.

## Acknowledgements

We are grateful for help with initial MST experiments from Brian Kloss and Renato Bruni in the Center on Membrane Protein Production and Analysis at the New York Structural Biology Center, which receives funding from NIH grant P41GM116799. Electron microscopy was performed at the Cryo-EM core facility at NYU Langone Health. The authors acknowledge High Performance Computing core facility at NYU Langone Health as well as Research Computing at Arizona State University for providing computational and storage resources that have contributed to the research results reported within this paper. We also thank Bjørn Panella Pederson for critical reading of the manuscript.

Funding for this work was provided by NIH grants R01GM125081 (DL Stokes) and R35GM144109 (DL Stokes). A portion of this research was also supported by NIH grant U24GM129547 and performed at the PNCC at OHSU and accessed through EMSL (grid.436923.9), a DOE Office of Science User Facility sponsored by the Office of Biological and Environmental Research. MD simulations were performed using PSC Bridges at the Pittsburgh Supercomputing Center (allocation TG-MCB130177), a resource of the Extreme Science and Engineering Discovery Environment, which is supported by National Science Foundation grant ACI-1548562.

## Additional information

### Funding

| Funder | Grant reference number | Author |
| --- | --- | --- |
| National Institute of General Medical Sciences | R01GM125081 | David L Stokes |
| National Institute of General Medical Sciences | R35GM144109 | David L Stokes |
| National Institute of General Medical Sciences | U24GM129547 | David L Stokes |
| National Science Foundation | ACI-1548562 | Oliver Beckstein |

The funders had no role in study design, data collection and interpretation, or the decision to submit the work for publication.

### Author contributions

Adel Hussein, Shujie Fan, Data curation, Formal analysis, Investigation, Visualization, Methodology, Writing - original draft, Writing - review and editing; Maria Lopez-Redondo, Formal analysis, Investigation, Methodology; Ian Kenney, Software, Formal analysis, Methodology; Xihui Zhang, Investigation, Methodology; Oliver Beckstein, Conceptualization, Resources, Software, Supervision, Funding acquisition, Validation, Visualization, Methodology, Writing - original draft, Project administration, Writing - review and editing; David L Stokes, Conceptualization, Resources, Formal analysis, Supervision, Funding acquisition, Validation, Visualization, Methodology, Writing - original draft, Project administration, Writing - review and editing

### Author ORCIDs

Adel Hussein ⓘ http://orcid.org/0000-0001-6963-9118
Oliver Beckstein ⓘ http://orcid.org/0000-0003-1340-0831
David L Stokes ⓘ http://orcid.org/0000-0001-5455-8163

Reviewer #1 (Public Review): https://doi.org/10.7554/eLife.87167.3.sa1

Reviewer #2 (Public Review): https://doi.org/10.7554/eLife.87167.3.sa2
Reviewer #3 (Public Review): https://doi.org/10.7554/eLife.87167.3.sa3
Author Response https://doi.org/10.7554/eLife.87167.3.sa4

## Additional files

### Supplementary files
• MDAR checklist

### Data availability

Coordinates for the atomic models have been deposited in the Protein Data Bank with accession numbers 8F6E, 8F6F, 8F6H, 8F6I, 8F6J, 8F6K. The corresponding density maps have been deposited in the Electron Microscopy Data Bank (EMDB) with accession numbers EMD-28881, EMD-28882, EMD-28883, EMD-28884, EMD-28885, EMD-28886.The following datasets were generated. The raw simulation data are stored in OSF Project https://doi.org/10.17605/OSF.IO/Y8BA2 under the CC-By Attribution 4.0 International. All computer code is stored in GitHub repository https://github.com/Becksteinlab/yiip_analysis (copy archived at *Fan, 2023*) and archived in Zenodo at https://doi.org/10.5281/zenodo.8357618; code is published under the open source GNU General Public License v3.

The following datasets were generated:

| Author(s) | Year | Dataset title | Dataset URL | Database and Identifier |
|---|---|---|---|---|
| Shujie F, Oliver B | 2023 | MD simulations of the YiiP transporter protein | https://doi.org/10.17605/OSF.IO/Y8BA2 | Open Science Framework, 10.17605/OSF.IO/Y8BA2 |
| Fan S, Beckstein O | 2023 | Becksteinlab/yiip_analysis: 1.0.3 | https://doi.org/10.5281/zenodo.8357618 | Zenodo, 10.5281/zenodo.8357618 |
| Lopez-Redondo ML, Hussein AK, Stokes DL | 2022 | Cryo-EM structure of a Zinc-loaded wild-type YiiP-Fab complex | https://www.rcsb.org/structure/8F6E | RCSB Protein Data Bank, 8F6E |
| Lopez-Redondo ML, Hussein AK, Stokes DL | 2022 | Cryo-EM structure of a Zinc-loaded D51A mutant of the YiiP-Fab complex | https://www.rcsb.org/structure/8F6F | RCSB Protein Data Bank, 8F6F |
| Lopez-Redondo ML, Hussein AK, Stokes DL | 2022 | Cryo-EM structure of a Zinc-loaded asymmetrical TMD D70A mutant of the YiiP-Fab complex | https://www.rcsb.org/structure/8F6H | RCSB Protein Data Bank, 8F6H |
| Lopez-Redondo ML, Hussein AK, Stokes DL | 2022 | Cryo-EM structure of a Zinc-loaded symmetrical D70A mutant of the YiiP-Fab complex | https://www.rcsb.org/structure/8F6I | RCSB Protein Data Bank, 8F6I |
| Lopez-Redondo ML, Hussein AK, Stokes DL | 2022 | Cryo-EM structure of a Zinc-loaded D287A mutant of the YiiP-Fab complex | https://www.rcsb.org/structure/8F6J | RCSB Protein Data Bank, 8F6J |
| Lopez-Redondo ML, Hussein AK, Stokes DL | 2022 | Cryo-EM structure of a Zinc-loaded H263A/D287A mutant of the YiiP-Fab complex | https://www.rcsb.org/structure/8F6K | RCSB Protein Data Bank, 8F6K |
| Lopez-Redondo ML, Hussein AK, Stokes DL | 2023 | Cryo-EM structure of a Zinc-loaded wild-type YiiP-Fab complex | https://www.ebi.ac.uk/emdb/EMD-28881 | Electron Microscopy Data bank, EMD-28881 |
| Lopez-Redondo ML, Hussein AK, Stokes DL | 2023 | Cryo-EM structure of a Zinc-loaded D51A mutant of the YiiP-Fab complex | https://www.ebi.ac.uk/emdb/EMD-28882 | Electron Microscopy Data Bank, EMD-28882 |
| Lopez-Redondo ML, Hussein AK, Stokes DL | 2023 | Cryo-EM structure of a Zinc-loaded asymmetrical TMD D70A mutant of the YiiP-Fab complex | https://www.ebi.ac.uk/emdb/EMD-28883 | Electron Microscopy Data Bank, EMD-28883 |

| Author(s) | Year | Dataset title | Dataset URL | Database and Identifier |
|-----------|------|---------------|-------------|-------------------------|
| Lopez-Redondo ML, Hussein AK, Stokes DL | 2023 | Cryo-EM structure of a Zinc-loaded symmetrical D70A mutant of the YiiP-Fab complex | https://www.ebi.ac.uk/emdb/EMD-28884 | Electron Microscopy Data Bank, EMD-28884 |
| Lopez-Redondo ML, Hussein AK, Stokes DL | 2023 | Cryo-EM structure of a Zinc-loaded D287A mutant of the YiiP-Fab complex | https://www.ebi.ac.uk/emdb/EMD-28885 | Electron Microscopy Data Bank, EMD-28885 |
| Lopez-Redondo ML, Hussein AK, Stokes DL | 2023 | Cryo-EM structure of a Zinc-loaded H263A/D287A mutant of the YiiP-Fab complex | https://www.ebi.ac.uk/emdb/EMD-28886 | Electron Microscopy Data Bank, EMD-28886 |

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
