## [Editor Report · eLife assessment]

This **important** and elegant study uses experimental structural data, ion affinity measurements, and computational methods to provide insight into the thermodynamic landscape of cation transporters of the Cation Diffusion Facilitator (CDF) superfamily, together with a detailed structural investigation of the role of the three zinc(II) binding sites of the YiiP family member. Overall, the support for the proposed transport cycle of YiiP is **compelling**. This work will be of interest to biologists and biophysics who work with membrane transporters.

---

## [Referee Report · Reviewer #1 (Public Review)]

The manuscript by Hussein et al. uses cryoEM structure, microscale thermophoresis (MST), and molecular dynamics simulations (conventional and CpHMD) to unravel the Zn2+ and proton role in the function of the Cation Diffusion Facilitator YiiP. First, they generate mutants that abolish each of the three Zn2+ models to study the role of each of them separately, both structurally and functionally. Next, they used a Monte Carlo approach refining the CpHMD data with the MST points to establish the Zn2+ or proton binding state depending on the pH. That predicted a stoichiometry of one Zn2+ to 2 or 3 protons (1:3 under lower pH values). Finally, they proposed a mechanism that involves first the binding of Zn2+ to one low-affinity site and then, after the Zn2+ migrates to the highest affinity site in the transmembrane portion of the protein. The lack of Zn2+ in the low-affinity site might induce occlusion of the transporter.

The manuscript is well-written it is of interest to the field of Cation Facilitator Transporters. It is also an excellent example of a combination of different techniques to obtain relevant information on the mechanism of action of a transporter.

---

## [Referee Report · Reviewer #2 (Public Review)]

In this work, the authors reported cryo-EM structures of four types of zinc-binding site mutants of a bacterial Zn2+/H+ antiporter YiiP, and proposed distinct structural/functional roles of each of the binding sites in the intramolecular Zn2+ relay and the integrity of the homodimeric structure of YiiP. MST analysis using the mutants with a single Zn2+-binding site at different pH further clarified the pH dependence of Zn2+ binding affinity of each site. Moreover, the inverse Multibind approach refined the CpHMD pKa values of the key Zn2+-binding residues so that they agreed with the MST data. Consequently, energetic coupling of Zn2+ export to the proton-motive force has been suggested. These findings definitely provide new mechanistic insight into this Zn2+/H+ antiporter.

---

## [Referee Report · Reviewer #3 (Public Review)]

This contribution focuses on the zinc(II) transporter YiiP, a widely used model system of the Cation Diffusion Facilitator (CDF) superfamily. CDF proteins function as dimers and are typically involved in the maintenance of homeostasis of transition metal ions in organisms from all kingdoms of life. The system investigated here, YiiP, is a prokaryotic zinc(II)/H+ antiporter that exports zinc(II) ions from the cytosol. The authors addressed multiple crucial questions related to the functioning of YiiP, namely the specific role of the three zinc(II) binding sites present in each protomer, the zinc(II):H+ stoichiometry of antiport, and the impact of protonation on the transport process. Clarity on all these aspects is required to reach a thorough understanding of the transport cycle.

The experimental approach implemented in this work consisted of a combination of site-directed mutagenesis, high-quality 3D structural determination by cryoEM, microscale electrophoresis, thermodynamic modeling and molecular dynamics. The mutants generated in this work removed one (for the structural characterization) or two (for microscale electrophoresis) of the three zinc(II) binding sites of YiiP, allowing the authors to unravel respectively the structural role of metal binding at each site and the metal affinity of every site individually. pH-dependent measurements and constant pH molecular dynamics simulations, together with the metal affinity data, provided a detailed per-site overview of dissociation constants and Ka values of the metal-binding residues, casting light on the interplay between protonation and metal binding along the transport cycle. This thermodynamic modeling constitutes an important contribution, with consistent experimental information gained from the various mutants.

Overall the authors were successful in providing a model of the transport cycle (Figure 5) that is convincing and well supported by the experimental data. The demonstration that two protomers act asymmetrically during the cycle is another nice achievement of this work, confirming previous suggestions. This novel overview of the cycle can constitute a basis for future work on other systems such as human ZnT transporters, also exploiting a methodological approach for the thermodynamic of these proteins similar to the one deployed here. The latter approach may be applicable also to other superfamilies of metal transporters.

---

## [Author Response]

The following is the authors’ response to the original reviews.

Thank you for reviewing our manuscript " Energy Coupling and Stoichiometry of Zn2+/H+ Antiport by the Cation Diffusion Facilitator YiiP". After carefully considering the reviewer's comments, we have made substantial changes to the manuscript, which we believe is now much improved. In addition to clarifying various points raised by the reviewers, we have also added a variety of new data from both experimental and computational studies. We hope that these changes will satisfy the reviewers such that we can move forward towards finalizing the publication process.

New data added to this revision includes

• SEC profiles comparing D287A and D287/H263A before and after complex formation with Fab to illustrate formation of higher order oligomerization (Suppl. Fig. 6).

• Control trace from MST using Mg2+ to illustrate reproducibility (Suppl. Fig. 6).

• Results from MD simulation of D72A mutant to explore the Asp72-Arg210 salt bridge as a stabilizing element (Fig. 4)

• Analysis of cavities in WT and D70A_asym structures to illustrate occlusion of site A (Suppl. Fig. 13).

• In addition, we have redone MD simulations for YiiP with site B empty. These simulations were originally done (3 x 1 μs) with a modified version of the zinc dummy model and we have redone them (6x1 μs) using our previously published zinc dummy model to be strictly consistent with other simulations on holo, apo and D72A structures. The new results are qualitatively consistent with the previous simulations and our conclusions remain unchanged.

In addition, the text has been modified and several figures have been updated to address concerns of the reviewers as described below.

Although figures will ultimately be renumbered to conform with eLife formatting, they have retained their original numbers for this revision to prevent confusion, except that Suppl. Fig. 13 is a new figure added at the request of reviewer 2.

**Reviewer #1 (Recommendations For The Authors):**
I have only a few comments that might need clarification from the authors:If the unbinding of Zn2+ to site B triggers the occlusion (and maybe the OF state) and the external pH does not affect that binding, how is it prevented from being always bound to Zn2+ and thus occluded also while it should be transporting protons (B to C panels in Figure 5)? Are there some other factors that I am missing?

Our data shows that the affinity of site B is low (micromolar), especially relative to the concentration of free Zn within the cytosol (picomolar - nanomolar). Therefore, we would expect that site B is normally empty and that the resting state would be represented by panel D in Figure 5. An elevation of Zn concentration, or delivery of Zn to the transporter by some as yet uncharacterized binding protein, would initiate the cycle starting with panel E.

It is notable that the TM2/TM3 loop adopts a novel conformation in the occluded state, in which it extends to interact with the CTD (panel G in Figure 5). In this conformation, the Zn binding site is disrupted, thus preventing binding of additional Zn ions to the TM2/TM3 loop. Although we do not know how this loop behaves as the protein transitions to the outward-facing state (panels A & B), it is tempting to speculate that it retains the extended conformation until the protein returns to the inward-facing, resting conformation in panel D. This idea has been added to the revised manuscript (line 464).

In addition, we have added a sentence (line 507) to explicitly state our assumption that Zn only binds to site B in the IF state.

I am not an expert on experiments, but the results for mutants that abolish site C are difficult to understand. For D287A/H263A, the SEC columns data suggest a population of higher oligomers. Still, for the D70A/D287A/H263A and D51A/D287A/H263A, they showed a native dimer. I understand your suggestion that the Fab induces the domain swap, but how do you explain the double mutant SEC column result? Please elaborate.

The unexpected behavior of site C mutants certainly introduces complexity into our study. Considering all the ins and outs of our analyses, we are confident that site C is a high-affinity site that is constitutively occupied and serves as a structural site to stabilize the architecture of the native homodimer. In the original submission, we included SEC profiles for D287A and D287A/H263A in Suppl. Fig. 4 as well as profiles for D70A/D287A/H263A and D51A/D287A/H263A in Suppl. Fig. 6. The former in Suppl. Fig. 4 characterize the complex between mutant YiiP and Fab (for cryo-EM), whereas the latter in Suppl. Fig. 6 represent YiiP in the absence of Fab (for MST). In the absence of Fab, the mutations do not alter the elution volume at ~12 ml, consistent with the conclusion that the native YiiP homodimer remains unperturbed. In the presence of Fab, mutations affect the SEC profile in two ways: a shift in the main peak to ~11 ml, and appearance of a subsidiary peak at ~10 ml. The shift of the main peak can be explained by formation of a complex between YiiP and Fab. Presence of the subsidiary peak - seen for D70A, D287A, and D287A/H263A mutants - can be explained by formation of a dimer of dimers (4 YiiP + 4 Fab), which could be isolated as a subpopulation of particles during the processing of cryo-EM images. For D70A and D287A, the individual dimers were unperturbed in this dimer-of-dimers. In fact, we used masking and signal subtraction to isolate the individual dimers and included them in the final reconstruction together with the more prevalent dimeric species (2 YiiP + 2 Fab).

The D287A/H263A-Fab complex behaved differently. The main peak of the SEC profile was shifted to 10 ml, indicating that a dimer of dimers was the prevalent complex; absence of a peak at 11 ml indicated that isolated dimeric complexes were no longer present in the solution. Furthermore, the subsidiary peak was at ~9 ml, indicating an even larger complex not seen in the other preparations. The appearance of particles in cryo-EM images were distinct from the other mutants (e.g., compare 2D classes shown in panels C and D in Suppl. Fig. 4). 3-D structures revealed dimer-of-dimers with the domain swap as well as larger linear oligomers. Although not well resolved due to preferred orientation, it appears that these linear oligomers consist of a propagated domain swap.

We have included some new data to bolster our conclusion that, although the D287A/H263A mutant destabilized site C, Fab binding was responsible for inducing the domain swap. The new data, presented in Suppl. Fig. 6, shows an SEC profile for a preparation of D287A/H263A both before and after formation of the complex with Fab. In addition to including this new data, we have amplified our description of these SEC profiles under the heading "Zn2+ binding affinity" in the paragraph starting on line 289 to try to clarify this complex issue for the reader.

Since in the D287A mutant, you are disrupting the preferred tetrahedral coordination of Zn2+, but it still binds, do you observe any waters that compensate for the missing aspartate? Maybe in the MD simulations?

Unfortunately, the resolution of the cryo-EM maps are not high enough to resolve water molecules that we assume are present at sites B and C. For the MD simulations, we did not use mutants, but simply removed Zn from each of the sites. So we are unfortunately not able to answer this question with the available data.

**Reviewer #2 (Recommendations for The Authors):**
1. It is no doubt that cryo-EM structures of four types of zinc-binding site mutants of a bacterial Zn2+/H+ antiporter YiiP provide important insight into distinct structural/functional roles of each of the binding sites. However, overall resolution of the cryo-EM maps presented in this paper is not high enough to address the Zn2+ coordination structures, the kinked TM5 segment seen in a D51A mutant, and the extended conformation of TM2/TM3 loop seen in the D70A asymmetric dimer. It would be better to highlight the density of the above regions and discuss the vitality of their structure models. Similarly, the presence of additional water molecules at sites B and C (line 117) do not seem convincing.

We are completely sympathetic with the recommendation of illustrating the map quality as thoroughly as possible. We hope that interested readers will download map and model from the respective PDB and EMDB repositories and see for themselves. Nevertheless, we have provided several new figure panels to illustrate explicitly the densities associated with the kinked TM5 segment in the D51A mutant (Suppl. Fig. 2) and the extended TM2/TM3 loop in the D70A mutant (Suppl. Fig. 5) and have referred to them at appropriate places in the text (line 128 and line 151). In Suppl. Fig. 5, we also included figure panels to show densities for this loop in WT and D287A/H263A mutants.

It is true that the maps are generally of insufficient resolution to clearly define the coordination of Zn. The relevant densities are shown for all sites in all mutants in Suppl. Fig. 2. Despite this shortcoming, the coordination geometry is well established by the previous, higher resolution X-ray crystal structure as well as by MD simulations. Each site is shown in the insets of Fig. 1b, c and d. The new cryo-EM densities and resulting models are consistent with this coordination, which we have now pointed out in the legend to Fig. 1. The important point is that the new cryo-EM maps document the occupancy of ions at the individual sites as well as the large scale conformational changes associated with this occupancy, which was the main goal of the study.

Finally, we agree that the presence of additional water molecules at the sites is not well supported; because this issue has little bearing on our analysis, these comments have been removed.

1. Identification of the occluded state in D70A asymmetric dimer is exciting, hence this reviewer recommends the authors to highlight the structure of this state more effectively in comparison with the IF/OF states. It would be better to show the side views of the superimposition between the occluded and IF/OF states, and the pore profile and radius in the TM domain of these three states. The authors should also show the density map of site A (including M2 and M5) in the occluded protomer of the asymmetric dimer in Suppl. Fig. 2. Additionally, the authors should include information regarding the cytosolic or periplasmic view in the legend of Figure 3A, B, D, F, G, and H.

As suggested, we have prepared a new supplemental figure juxtaposing the IF and occluded states and depicting differences in pore radius and accessibility of site A (Suppl. Fig. 13, initially referred to on line 152 and various other locations in the manuscript with methods described on line 680). However, we unfortunately do not have a structure in the OF state to complete this comparison.

The density map for site A including M2 and M5 of the occluded protomer is shown in Suppl. Fig. 2 in which density thresholds have been adjusted to show the helices.

We have updated the figure legend for Figure 2 (referred to as Figure 3 by the reviewer) with the orientation of view, which are all from the cytoplasm looking toward the membrane.

1. MST analyses using the YiiP mutants with a single Zn2+-binding site at different pH are useful, and the data interpretation in combination with computational approaches of CpHMD and MST inference are nice challenges, indeed. However, it may, in a sense, appear that the MD simulations have been carried out intentionally and/or forcibly so that the outcomes are compatible with the experimental MST data. Although this is not unusual or unacceptable, this reviewer is concerned that the determined pKa values of some residues, especially Asp residues at Site A, are unusually high. The validity of this outcome should be discussed from physicochemical viewpoint; what factors raise the pKa of Asp51 and Asp159 so high. In this context, the MST inference titration curve seems unusually steep for D159 (and H155), of which validity needs to be discussed. This reviewer is also concerned about the large variations per measurement in the MST experiments (Suppl. Fig 6 E, F, and G). Are such large variations common to this experiment? Optimization of the measurement conditions such as protein concentration, and/or increase of AlexaFluor-488 labeling efficiency might greatly improve the reproducibility per measurement. The authors should include information on which residue(s) is labeled with AlexaFluor 488 in YiiP (line 641).

One of the outcomes of our so-called MST-inference algorithm was the conclusion that protonation states for H155 and D159 were coupled. The basis for this conclusion is described in some detail in the Methods section (paragraph starting on line 1025) and results in cooperativity in the protonation state of these two residues. This cooperativity explains the unusually steep binding curve in Suppl. Fig. 10e. We added a couple of sentences to explain this result in the Results under "Zn2+ binding affinity", line 352.

There is indeed precedent for increased pKas of acidic residues based on experimental measurements for Glu and inferred for Asp, both in membrane proteins. Computational approaches similar to the ones we use (including some of our own earlier work) have also pointed to elevated pKas by 1-3 units for Asp residues. We included a paragraph in the Discussion of Stoichiometry and energy coupling (line 537) citing these references and explaining that such pKa shifts reflect strong Coulomb interactions of titratable residues in close proximity in the low dielectric environment of the membrane.

We believe there is a misunderstanding about our presentation of raw data for the MST experiments in Suppl. Fig. 6. Panels E, F and G show an overlay of data from the entire Zn titration, which is therefore expected to change according to the Zn concentration in each capillary. We have revised the corresponding legend to clarify the plots. We have also included traces from a Mg2+ titration as a negative control that better illustrates the reproducibility of these measurements.

The AlexaFluor dye contained the reactive NHS group which preferentially targets the N-terminus of the polypeptide chain. Although labeling of lysine side chains is possible, we do not expect much given the low labeling stoichiometry of ~1:1 used for our experiments. We updated the Methods section under MST experiments (line 689) with this information.

**Reviewer #3 (Recommendations For The Authors):**
By measuring the binding affinity of site A using the D70A mutant that retains site C at pH 5.6 is should be possible to verify if the affinity reported in Table 2 is affected by the quaternary structure of the system. The 40-fold difference in affinity between site A and site C at pH 5.6 should be sufficiently large to permit a meaningful measurement.

To address this suggestion, we have included additional data in Table 2 from the D70A/D287A mutant. Based on the cryo-EM structure of D287A, we expect that site C is still intact, which is why it was omitted from the original manuscript. However, the affinities measured at pH 6 and 7 are very consistent with those from the triple mutant (D70A/D287A/H263A), supporting the idea that complete abolishment of site C does not affect measurement of affinity at sites A or B. This additional data is presented in the section on "Zn2+ binding affinity" on line 304. We also note that the SEC profiles in the absence of Fab are consistent with formation of the native homodimer for all the mutants, as described in our response to reviewer 1 and now shown in Suppl. Fig. 6.

More details should be provided on the force field used for zinc(II) ions in MD simulations. Currently, there is only a reference to another article, where this info is in the caption of a supplementary figure.

We added a summary of our previous work to develop a non-bonded dummy model for Zn(II) on line 727 in the Methods section entitled "Overview of the MD simulations. However, we would like to point out that all details on the parameter development and the parameters themselves are stated in the Methods section “Classical force field model for Zn(II) ions” in our previous paper [Lopez-Redondo et al, J Gen Physiol 143 (2021)] and parameter files are available as package 2934 in the Ligandbook repository https://ligandbook.org/package/2934 .

We also realized that in the originally submitted version of this manuscript we reported “empty site B” simulations with an updated and experimental non-bonded Zn(II) dummy model that has close to experimental first-solvation shell water residence times but slightly worse solvation free energy. Although that does not really matter for these simulations because there was no Zn2+ ion in site B, we nevertheless performed a new set of 6 x 1 µs simulations with our published (J Gen Physiol 2021) Zn(II) model to make all simulations fully consistent with each other. The results remained qualitatively the same, with a lack of zinc ions in site B leading to increased flexibility in the TM2/3 loop and ultimately destabilization of the TMD-CTD interaction.